# GROUPED DIRICHLET DIFFUSION FOR STRUCTURED GENERATIVE MODELING

## ABSTRACT

We present Grouped Dirichlet Diffusion (GDD), a novel generative model that employs the Grouped Dirichlet distribution to facilitate hierarchical and structured diffusion processes for high-dimensional bounded probability vectors, such as multichannel images. Unlike conventional diffusion methods that rely on Gaussian noise, GDD partitions data into meaningful feature groups (e.g., color channels in images) to preserve intra-group dependencies while allowing adaptive inter-group interactions over diffusion timesteps. Our theoretical framework ensures that both the forward marginals and reverse conditionals remain within the Grouped Dirichlet family, enabling closed-form transitions through multiplicative noise scheduling. This approach not only simplifies training dynamics but also guarantees numerical stability during sampling. Additionally, we replace the traditional evidence lower bound (ELBO) with a loss function based on the Kullback-Leibler divergence. Experimental evaluations validate the feasibility of GDD, with quantitative metrics that demonstrate superior image generation performance compared to traditional diffusion models and several contemporary image generation methods.

## 1 INTRODUCTION

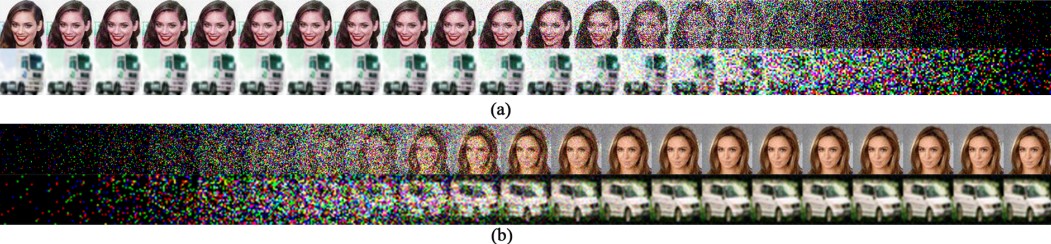

Figure 1: (a) Forward Diffusion and (b) Reverse Diffusion in Grouped Dirichlet Diffusion. This figure illustrates the hierarchical diffusion process: the forward pass progressively corrupts high-dimensional bounded probability vectors—such as multichannel image data—while maintaining intra-group dependencies, and the reverse pass reconstructs the original data through adaptive inter-group interactions.

Generative models (Vahdat et al., 2021; Wang et al., 2024) have garnered widespread attention in machine learning and statistics in recent years, with diffusion-based methods emerging as particularly powerful. These models excel in tasks such as image generation, restoration, synthesis (Gu et al., 2022; Oguz et al., 2024), 2D-to-3D transformation (Luo & Hu, 2021; Poole et al., 2023), reinforcement learning (Black et al., 2024), and video synthesis (Shi et al., 2024). Their development has significantly advanced the modeling of complex distributions, evolving from unconditional image generation Menick & Kalchbrenner (2019) to the incorporation of conditional information for classification and regression tasks. For instance, the CARD model (Han et al., 2022) effectively predicts multimodal conditional distributions by integrating denoising diffusion mechanisms with pre-trained conditional mean estimators. Diffusion methods have also demonstrated value in diverse domains, including antigen-specific antibody design (Huang et al., 2024; Luo et al., 2022), image density estimation, and speech synthesis flow modeling (Gao et al., 2020).

However, despite these advances, traditional diffusion methods based on Gaussian (Ho et al., 2020; Guo et al., 2023) or beta distributions (Zhou et al., 2023) are not designed to explicitly model group-level dependencies or hierarchical group–element structures in high-dimensional bounded data, such as multichannel images. This limitation curbs the ability of such models to represent the nuanced interrelationships among different data channels effectively.

Motivated by these challenges, we introduce Grouped Dirichlet Diffusion (GDD), a novel diffusion-based generative framework designed to overcome these shortcomings. By leveraging the grouped Dirichlet distribution, GDD partitions data into meaningful feature groups (e.g., color channels in images), preserving intra-group dependencies while dynamically adapting inter-group interactions over time (Lee et al., 2023). This structured approach enhances modeling flexibility and improves numerical stability by ensuring that the diffusion process consistently operates within the well-defined simplex constraints of the Dirichlet distribution.

Diffusion-based generative models have evolved notably with approaches like Gaussian diffusion—which gradually transforms images into Gaussian noise and then reverses the process to generate realistic images—and beta diffusion, which employs scaled and shifted beta distributions. Beta diffusion integrates demapping and denoising techniques, uses multiplicative transformations in both its forward and reverse processes, and introduces Kullback–Leibler divergence upper bounds (KLUBs) as a more effective optimization criterion than traditional negative evidence lower bounds (ELBOs), resulting in more stable training dynamics. Building on these developments, GDD extends traditional methods by incorporating grouped dirichlet distributions that are specifically tailored for high-dimensional, structured data (Weilbach et al., 2023).

The construction of the GDD follows a three-step procedure that effectively captures group-wise dependencies:

**Forward Diffusion Process:** The data is partitioned into several groups and gradually corrupted by multiplicative noise following the Dirichlet distribution. Noise levels are progressively tuned to generate increasingly complex data distributions over time.

**Reverse Diffusion Process:** Given the corrupted data, the model denoises and reconstructs the original input by predicting the parameters of the grouped Dirichlet distribution, thereby restoring the data.

**Optimization:** Instead of employing traditional ELBO-based loss functions, the model uses KL divergence as the optimization criterion for both forward and reverse processes. Minimizing the KL divergence between the observed and generated data ensures that the model closely approximates the true data distribution at each step.

The primary contributions of this work are summarized as follows:

- **Introduction of GDD:** We propose GDD, a novel diffusion-based multiplicative generative model specifically designed for high-dimensional, bounded data.

- **Enhanced Modeling Flexibility:** Unlike traditional Gaussian-based diffusion models, GDD offers a more flexible and structured framework for capturing complex data patterns, particularly in multichannel images.

- **Innovative Loss Optimization:** We introduce the KL divergence upper bounds (KLUBs) as an effective loss objective and extend this approach by formulating the KL divergence for grouped Dirichlet distributions using log-beta divergence, which reinforces the theoretical foundation of our model and enhances its performance.

In addition to these innovations, we detail the implementation of grouped dirichlet diffusion by defining the grouped dirichlet distribution and elucidating its connection to the beta distribution. We describe both the forward and reverse diffusion processes and illustrate how, in our framework, the marginal and conditional distributions of each group follow scaled and shifted Dirichlet laws—akin to beta diffusion. We illustrate the forward grouped dirichlet diffusion process in Figure 1 (a), which simultaneously adds noise to the data, and the reverse one in Figure 1 (b). This compatibility with the hierarchical structure of grouped probability vectors is maintained through logit-space operations. We further discuss the network architecture, the design of time-step-related decay coefficients, and the computation of the KL divergence for multiple grouped dirichlet distributions.

Collectively, these contributions enable GDD to deliver superior performance in generating high-dimensional, structured data, thereby opening new avenues for image synthesis and other complex generative tasks.

## 2 RELATED WORK

Early diffusion generative models focused on probabilistic frameworks (Chen et al., 2024a; Kim et al., 2022; Lawson et al., 2019; Cao et al., 2024) and improvements such as noise scheduling and sampling strategies to enhance image and speech synthesis quality (Chen et al., 2024b; Kim et al., 2023; Bartosh et al., 2024; Austin et al., 2021; Franceschi et al., 2023; Zhang et al., 2024). For example, (Karras et al., 2022) decoupled sampling and designed a pre-modulation module, improving efficiency and stability. Despite progress, modeling high-dimensional data and uncertainty remains challenging. Recent works address this by integrating structured priors with nonparametric Bayesian methods, notably Dirichlet diffusion (Avdeyev et al., 2023b; Ongaro & Migliorati, 2013). (Knowles et al., 2011) used Dirichlet diffusion trees with ARD priors for dimensionality reduction and group extraction. (Ruggiero, 2014) analyzed species diversity via Poisson-Dirichlet diffusion. Advances also include message passing for approximate inference (Knowles et al., 2011) and fast Dirichlet flow generation (Stärk et al., 2024b). We propose GDD, extending beta diffusion with grouped Dirichlet distributions (Ng et al., 2008). GDD partitions data into semantic groups, preserving intragroup dependencies and enabling adaptive intergroup interactions. It provides closed-form estimators and reduces latent variables, enhancing estimation efficiency. Experiments show faster convergence than traditional methods. These developments improve diffusion models' efficiency and expressiveness, broadening applicability to complex data. GDD combines structured priors with diffusion, advancing generative modeling in theory and practice.

## 3 METHODOLOGY

In this section, we detail our approach to implementing Grouped Dirichlet Diffusion. We begin by defining the grouped dirichlet distribution and clarifying its differences and connections to the beta distribution. We then describe the forward grouped dirichlet diffusion process alongside the reverse process. In our framework, both the marginal and conditional distributions of each group follow scaled and shifted Dirichlet laws—akin to beta diffusion—ensuring compatibility with the hierarchical structure of grouped probability vectors while maintaining numerical stability through logit-space operations. Additionally, we discuss the network architecture and the design of time-step-related decay coefficients. Finally, we elaborate on the loss function design, explaining how we optimize with KL divergence and compute it for multiple grouped dirichlet distributions.

### 3.1 BASIC DEFINITION

Let $G$ denote the number of independent groups. For each group $g \in \{1, \ldots, G\}$, $\mathbf{x}_g = (x_{g1}, x_{g2}, \ldots, x_{gK})$ is the random vector, it lies on the $K$-dimensional simplex, meaning it satisfies: $\sum_{i=1}^{K} x_{gi} = 1$, $x_{gi} \geq 0$ for all $i \in \{1, \ldots, K\}$. If each group $\mathbf{x}_g$ independently follows a Dirichlet distribution with parameter vector $\boldsymbol{\alpha}_g = (\alpha_{g1}, \ldots, \alpha_{gK})$, then the joint distribution of all groups is referred to as the grouped dirichlet Distribution. Formally, it is defined as: $p\left(\{\mathbf{x}_g\}_{g=1}^{G}; \{\boldsymbol{\alpha}_g\}_{g=1}^{G}\right) = \prod_{g=1}^{G} \text{Dir}\left(\mathbf{x}_g; \boldsymbol{\alpha}_g\right)$. The probability density function (PDF) for a single group's Dirichlet distribution is given by: $\text{Dir}\left(\mathbf{x}_g; \boldsymbol{\alpha}_g\right) = \frac{1}{B(\boldsymbol{\alpha}_g)} \prod_{i=1}^{K} x_{gi}^{\alpha_{gi}-1}$, where $B(\boldsymbol{\alpha}_g)$ is the multivariate beta function that serves as the normalization constant: $B(\boldsymbol{\alpha}_g) = \frac{\prod_{i=1}^{K} \Gamma(\alpha_{gi})}{\Gamma\left(\sum_{i=1}^{K} \alpha_{gi}\right)}$, where, $\Gamma(\cdot)$ denotes the gamma function.

For the Grouped Dirichlet distribution (Ng et al., 2008), the joint PDF across all $G$ groups is the product of individual group densities: $p\left(\{\mathbf{x}_g\}; \{\boldsymbol{\alpha}_g\}\right) = \prod_{g=1}^{G} \left[\frac{1}{B(\boldsymbol{\alpha}_g)} \prod_{i=1}^{K} x_{gi}^{\alpha_{gi}-1}\right]$. In logarithmic form, this expression becomes:

$$\ln p\left(\{\mathbf{x}_g\}; \{\boldsymbol{\alpha}_g\}\right) = \sum_{g=1}^{G} \left[-\ln B(\boldsymbol{\alpha}_g) + \sum_{i=1}^{K} (\alpha_{gi} - 1) \ln x_{gi}\right]. \tag{1}$$

The Grouped Dirichlet distribution generalizes the Dirichlet to hierarchically structured data partitioned into $G$ independent groups. Group $g$ has concentration vector $\boldsymbol{\alpha}^{(g)}$ that generates a sub-simplex $\mathbf{x}^{(g)}$; concatenating all groups forms $\mathbf{x} \in \Delta_{K-1}$. For $K = 2$ each group reduces to a Beta law, yielding $G$ independent Betas. Because Dirichlet and Beta are members of the exponential family, they support efficient maximum-likelihood and Bayesian estimation with uniquely optimal divergence objectives. Standard Gaussian (Ho et al., 2020) or scalar-Beta diffusion (Zhou et al., 2023) models violate simplex non-negativity, unit-sum constraints, and overlook group dependencies. Grouped Dirichlet Diffusion (GDD) replaces each scalar Beta with a Dirichlet, clusters correlated channels, and employs a shared noise schedule. This design preserves the simplex, captures intra-/inter-group structure, retains exponential-family tractability, and provides closed-form forward marginals, analytic reverse dynamics, and an efficient KL objective—while requiring only minor changes to existing diffusion pipelines.

## 3.2 FORWARD GROUPED DIRICHLET DISTRIBUTION

The forward process of Grouped Dirichlet Diffusion gradually perturbs structured data (e.g., multi-group probability vectors) toward a predefined prior (e.g., a uniform Dirichlet) through a series of noise-corrupted steps.

Let $\mathbf{x}_0 = \{\mathbf{x}_g\}_{g=1}^G$ denote the observed data, where each group $\mathbf{x}_g$ lies on a $K$-dimensional simplex:

$$\sum_{i=1}^K x_{gi} = 1 \quad \text{and} \quad x_{gi} \geq 0 \quad \text{for all } i \in \{1, \ldots, K\}. \tag{2}$$

Let $\mathbf{z}_s$ and $\mathbf{z}_t$ represent the corrupted versions of $\mathbf{x}_0$ at times $s$ and $t$ (with $0 < s < t < 1$). The forward diffusion process involves sampling from the marginal distribution $q(\mathbf{z}_t \mid \mathbf{x}_0)$ at any time $t$ and obtaining analytical expressions for the conditional distribution $q(\mathbf{z}_t \mid \mathbf{z}_s, \mathbf{x}_0)$ when $s < t$.

In the forward Grouped Dirichlet diffusion chain, diffusion scheduling parameters $\alpha_t$ control the decay of the expected values across groups. Specifically, for each group $g$:

$$\mathbb{E}[\mathbf{z}_{g,t} \mid \mathbf{x}_{g0}] = \alpha_t \, \mathbf{x}_{g0}, \tag{3}$$

where $\alpha_t$ monotonically decreases from $\alpha_0 \approx 1$ (near $t = 0$) to $\alpha_1 \approx 0$ (near $t = 1$). A concentration parameter $\eta > 0$ governs the dispersion of the noise around this expected value; higher $\eta$ yields a tighter distribution around $\alpha_t \, \mathbf{x}_0$, while lower $\eta$ increases entropy, mimicking uniform noise.

The noise level $\alpha_t$ is computed via a sigmoid-based nonlinear schedule. Let $T$ denote the total number of diffusion steps. For each timestep $k \in \{0, 1, \ldots, T\}$, define the normalized time $t_k = \frac{k}{T}$, $t_k \in [0, 1]$. A logit-transformed parameter $\tilde{\alpha}_k$ is computed using power-law interpolation between two constants $c_{\text{start}}$ and $c_{\text{end}}$:

$$\tilde{\alpha}_k = c_{\text{start}} + (c_{\text{end}} - c_{\text{start}}) \cdot t_k^\gamma, \tag{4}$$

where $c_{\text{start}}$ and $c_{\text{end}}$ set the initial and final logit values, and $\gamma$ determines the interpolation curvature. The final scheduling parameter is obtained via the sigmoid function:

$$\alpha_k = \sigma(\tilde{\alpha}_k) = \frac{1}{1 + e^{-\tilde{\alpha}_k}}. \tag{5}$$

This formulation ensures that $\alpha_k$ smoothly transitions from 1 to 0 over the diffusion process, balancing noise injection with data preservation. Compared to traditional linear or cosine schedules, the sigmoid-based schedule provides finer control over intermediate timesteps(particularly when $t_k$ is near 0 or 1), which has been observed to offer greater flexibility than the linear schedule for image generation. This schedule bears resemblance to the sigmoid-based one introduced for Gaussian diffusion (Kingma et al., 2021; Jabri et al., 2023).

To align the data with the model's dynamic range, the input $\mathbf{x}_0$ is transformed as follows:

$$\mathbf{x}_0 \leftarrow \mathbf{x}_0 \times S_{\text{scale}} + S_{\text{shift}}, \tag{6}$$

ensuring numerical stability during sampling and loss computation.

For each group $g$, corrupted samples $\mathbf{z}_{g,t_i}$ and $\mathbf{z}_{g,s_i}$ are drawn from a Dirichlet distribution parameterized by $\eta \alpha_{t_i} \mathbf{x}_{g0}$ and $\eta \alpha_{s_i} \mathbf{x}_{g0}$, respectively:

$$\mathbf{z}_{g,t_i} \sim \mathrm{Dir}\Big(\eta \alpha_{t_i} \mathbf{x}_{g0}\Big), \quad \mathbf{z}_{g,s_i} \sim \mathrm{Dir}\Big(\eta \alpha_{s_i} \mathbf{x}_{g0}\Big). \tag{7}$$

To enhance temporal coherence, a secondary timestep is defined as:

$$s_i = \pi \, t_i \quad (\pi \in (0,1)), \tag{8}$$

to compute $\alpha_{s_i}$, enabling the model to learn bidirectional transitions.

For the case $K = 2$ (i.e., when each group comprises two components), given the observed data $\mathbf{x}_0 = \{\mathbf{x}_{g0}\}_{g=1}^{G}$, where image channels are grouped as a prior modeling assumption to greatly simplify the mathematics and ensure closed-form marginals, the two univariate marginals are both distributed according to the grouped dirichlet distribution:

$$q(z_t \mid x_0) = \mathrm{GDD}\big(\eta \, \alpha_t \, x_0, \, \eta \, (1 - \alpha_t \, x_0)\big), \tag{9}$$

$$q(z_s \mid x_0) = \mathrm{GDD}\big(\eta \, \alpha_s \, x_0, \, \eta \, (1 - \alpha_s \, x_0)\big). \tag{10}$$

Following the methodology of Beta Diffusion (Zhou et al., 2023), for each group $g$, the corrupted sample $\mathbf{z}_{g,t}$ is generated by scaling the intermediate sample $\mathbf{z}_{g,s}$ with a Grouped Dirichlet-distributed variable $\boldsymbol{\pi}_{g,s \to t}$:

$$\mathbf{z}_{g,t} = \mathbf{z}_{g,s} \odot \boldsymbol{\pi}_{g,s \to t}, \quad \boldsymbol{\pi}_{g,s \to t} \sim \mathrm{Dir}\Big(\eta \, \alpha_t \, \mathbf{x}_{g0}, \, \eta \big(\alpha_s - \alpha_t\big)\mathbf{x}_{g0}\Big), \tag{11}$$

where $\odot$ denotes element-wise multiplication under simplex constraints, and $\mathbf{z}_{g,s} \sim q(\mathbf{z}_s \mid \mathbf{x}_0)$ is sampled from the marginal distribution at time $s$. The specific algorithm is implemented in Algorithm 1.

### 3.3 REVERSE GROUPED DIRICHLET DISTRIBUTION

We extend the framework of Gaussian and Beta diffusion by utilizing the conditional distribution $q(\mathbf{z}_s \mid \mathbf{z}_t, \mathbf{x}_0)$ to define the reverse process $p_\theta(\mathbf{z}_s \mid \mathbf{z}_t)$. In constructing the reverse Grouped Dirichlet diffusion chain, our goal is to learn transitions from $\mathbf{z}_t$ to $\mathbf{z}_s$ for $s < t$.

During inference, the original data $\mathbf{x}_0$ is not available; instead, it is approximated using a learned generator $f_\theta$, which predicts group-wise parameters $\{\hat{\boldsymbol{\alpha}}_g\}_{g=1}^{G}$ from the corrupted sample $\mathbf{z}_t$ and the timestep $t$:

$$\hat{\boldsymbol{\alpha}}_g = f_\theta(\mathbf{z}_t, t). \tag{12}$$

It represents the parameters for constructing the grouped Dirichlet distribution predicted by the model, as well as the image vector predicted by the model in practical operation. The reverse process then replaces $\mathbf{x}_{g0}$ with its approximation $\hat{\boldsymbol{\alpha}}_g$ and is defined as:

$$p_\theta(\mathbf{z}_s \mid \mathbf{z}_t) = \prod_{g=1}^{G} \mathrm{Dir}\Big(\mathbf{z}_{g,s}; \, \eta \, (\alpha_s - \alpha_t) \, \hat{\boldsymbol{\alpha}}_g, \, \eta \, \big(1 - \alpha_s \hat{\boldsymbol{\alpha}}_g\big)\Big). \tag{13}$$

For each group $g$, $\mathbf{z}_{g,s}$ is reconstructed from $\mathbf{z}_{g,t}$ by combining it with a grouped dirichlet perturbation $\mathbf{p}_{g,s-t}$:

$$\mathbf{z}_{g,s} = \mathbf{z}_{g,t} + \big(\mathbf{1} - \mathbf{z}_{g,t}\big) \odot \mathbf{p}_{g,s-t}, \tag{14}$$

$$\mathbf{p}_{g,s-t} \sim \mathrm{Dir}\Big(\eta \, (\alpha_s - \alpha_t) \, \mathbf{x}_{g0}, \, \eta \, \big(1 - \alpha_s \, \mathbf{x}_{g0}\big)\Big), \tag{15}$$

where $\mathbf{1}$ is a $K$-dimensional vector of ones, $\odot$ denotes element-wise multiplication under simplex constraints, and $\mathbf{z}_{g,t} \sim q(\mathbf{z}_t \mid \mathbf{x}_0)$.

To ensure numerical stability near simplex boundaries (e.g., when $x_{gi} \approx 0$), all Dirichlet sampling and parameter updates are performed in logit space $\mathrm{logit}(\mathbf{z}_{g,s})$:

$$\ln\Big(e^{\mathrm{logit}(\mathbf{z}_{g,t})} + e^{\mathrm{logit}(\mathbf{p}_{g,s-t})} + e^{\mathrm{logit}(\mathbf{z}_{g,t}) + \mathrm{logit}(\mathbf{p}_{g,s-t})}\Big), \tag{16}$$

where $\mathbf{p}_{g,s-t} \sim \mathrm{Dir}\Big(\eta\,(\alpha_s - \alpha_t)\,\hat{\boldsymbol{\alpha}}_g,\ \eta\,(1 - \alpha_s\,\hat{\boldsymbol{\alpha}}_g)\Big)$.

This alignment ensures: **1. Group Independence**: Transitions for each group $g$ are decoupled, preserving the hierarchical structure. **2. Simplex Constraints**: All operations adhere to the $K$-dimensional simplex, thereby preventing invalid probability vectors. The specific algorithm is implemented in Algorithm 2.

### 3.4 Loss Function Design

Since the Grouped Dirichlet distribution extends the Beta distribution, we adopt the KL Upper Bound (KLUB) originally proposed for Beta Diffusion (Zhou et al., 2023) — whose feasibility has been well established—and discuss its application to the multigroup Dirichlet distribution.

For a single group, consider two Dirichlet distributions defined over a $K$-dimensional simplex, $\mathrm{Dir}(x; \boldsymbol{\alpha})$ and $\mathrm{Dir}(x; \boldsymbol{\beta})$, where $\boldsymbol{\alpha} = (\alpha_1, \alpha_2, \ldots, \alpha_K)$, $\boldsymbol{\beta} = (\beta_1, \beta_2, \ldots, \beta_K)$. Let $\alpha_0 = \sum_{i=1}^{K} \alpha_i$, $\beta_0 = \sum_{i=1}^{K} \beta_i$. The KL divergence from $\mathrm{Dir}(x; \boldsymbol{\alpha})$ to $\mathrm{Dir}(x; \boldsymbol{\beta})$ is then given by: Let $\boldsymbol{\alpha} = (\alpha_1, \ldots, \alpha_K)$, $\boldsymbol{\beta} = (\beta_1, \ldots, \beta_K)$, and define $\alpha_0 = \sum_{i=1}^{K} \alpha_i$, $\beta_0 = \sum_{i=1}^{K} \beta_i$. The KL divergence between two Dirichlet distributions is defined as:

$$\mathrm{KL}\Big(\mathrm{Dir}(\boldsymbol{\alpha}) \,\|\, \mathrm{Dir}(\boldsymbol{\beta})\Big) = \int_{\Delta_{K-1}} p_{\boldsymbol{\alpha}}(\mathbf{x}) \ln \frac{p_{\boldsymbol{\alpha}}(\mathbf{x})}{p_{\boldsymbol{\beta}}(\mathbf{x})} \, \mathrm{d}\mathbf{x}, \tag{17}$$

where the simplex is defined as $\Delta_{K-1} = \Big\{ \mathbf{x} \in \mathbb{R}_{\geq 0}^K \,\Big|\, \sum_{i=1}^{K} x_i = 1 \Big\}$. For the Grouped Dirichlet distribution, define: $p = \prod_{g=1}^{G} \mathrm{Dir}(\boldsymbol{\alpha}_g), q = \prod_{g=1}^{G} \mathrm{Dir}(\boldsymbol{\beta}_g)$. The KL divergence between these two distributions is then:

$$D_{\mathrm{KL}}(p \| q) = \sum_{g=1}^{G} D_{\mathrm{KL}}\Big(\mathrm{Dir}(\boldsymbol{\alpha}_g) \,\|\, \mathrm{Dir}(\boldsymbol{\beta}_g)\Big). \tag{18}$$

We refer to this sum—equivalent to the Bregman divergence associated with the log-beta function—as the log-beta divergence. In practice, we must account for the discrepancy between $q(z_s \mid z_t)$ and $p_\theta(z_s \mid z_t)$. To mitigate error accumulation during time reversal, we integrate KLUB into the training objective, similar to the Beta diffusion framework. Specifically, the training objective minimizes a weighted combination of two KL upper bounds:

1. *Forward-Reverse KLUB*: Measures the divergence between the forward process $q(\mathbf{z}_{s_i} \mid \mathbf{z}_{t_i}, \mathbf{x}_0)$ and the reverse process $p_\theta(\mathbf{z}_{s_i} \mid \mathbf{z}_{t_i})$.

2. *Marginal KLUB*: Directly compares the corrupted sample $\mathbf{z}_{t_i}$ with the original data $\mathbf{x}_0$.

For the Grouped Dirichlet case, the total loss for sample $i$ is:

$$\mathcal{L}_i = \omega \sum_{g=1}^{G} D_{\mathrm{KL}}\Big(\mathrm{Dir}(\boldsymbol{\alpha}_{q,g}) \,\|\, \mathrm{Dir}(\boldsymbol{\alpha}_{p,g})\Big)$$
$$+ (1 - \omega) \sum_{g=1}^{G} D_{\mathrm{KL}}\Big(\mathrm{Dir}(\boldsymbol{\alpha}_{q,g}^{(*)}) \,\|\, \mathrm{Dir}(\boldsymbol{\alpha}_{p,g})\Big), \tag{19}$$

where $\boldsymbol{\alpha}_{q,g} = \eta\,\alpha_{s_i}\,\mathbf{x}_{g,0}$, $\boldsymbol{\alpha}_{p,g} = f_\theta(\mathbf{z}_{t_i}, t_i)$, $\boldsymbol{\alpha}_{q,g}^{(*)} = \eta\,\alpha_{t_i}\,\mathbf{x}_{g,0}$. We optimize the generator $f_\theta$ via stochastic gradient descent (SGD), and the hyperparameter $\omega$ controls the balance between the two loss components.

KLUB circumvents the need for exact computation of Gamma and digamma functions through the use of approximations or constrained forms, thereby reducing computational overhead. Moreover, executing operations in logit space (e.g., $\mathrm{logit}(\mathbf{z})$) enhances numerical stability by mitigating boundary effects near the simplex edges (e.g., when $x_{gi} \approx 0$). In addition, KLUB inherently provides smoother gradients compared to the exact KL divergence, reducing oscillations during optimization and accelerating convergence. The weighted design via $\omega$ enables flexible trade-offs between temporal coherence (enforced by forward-reverse transitions) and generation quality (ensured by marginal matching).

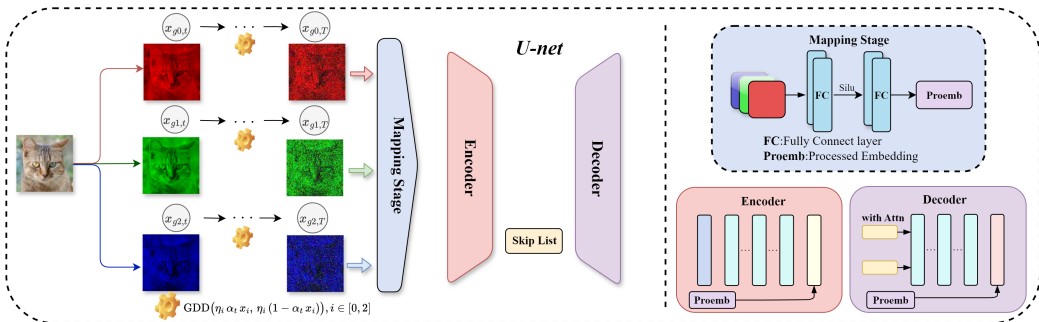

Figure 2: An overview of the proposed Grouped Dirichlet Diffusion (GDD) framework. The main pipeline adopts an encoder–decoder design, integrating a U-Net architecture with residual blocks, attention mechanisms, and skip connections. The encoder compresses grouped data (e.g., color channels in images) into a latent representation, while the decoder reconstructs or generates outputs by incorporating both global context (via attention) and fine-grained details (via skip connections). A separate mapping module uses fully connected layers and processed embeddings (Proemb) to align the learned representations with the Dirichlet diffusion process. Collectively, these components form a cohesive framework for capturing inter- and intra-group dependencies in high-dimensional bounded data.

## 3.5 NETWORK ARCHITECTURE

GDD employs a multi-stage network architecture that integrates hierarchical feature extraction with multi-scale information fusion (Song et al., 2021b), a proven approach in diffusion-based generative models. The framework consists of three main stages: mapping, encoding, and decoding, as illustrated in Figure 2.

**1. Mapping Stage**: The input noise, optionally combined with class or augmentation labels, is transformed into a latent embedding. Labels are encoded via positional or Fourier embeddings, followed by linear layers with SiLU activations. This embedding modulates subsequent network layers.

**2. Encoder (Downsampling) Stage**: Implemented as a `ModuleDict`, the encoder processes the input image through multiple resolution levels. An initial convolution maps input channels to a base number of feature maps. Subsequent levels apply UNetBlocks configured for downsampling, reducing spatial resolution. Auxiliary paths (e.g., skip or residual connections) preserve information across scales. Outputs at each level are saved as skip connections for the decoder.

**3. Decoder (Upsampling) Stage**: The decoder reverses the encoding process. Starting from the bottleneck, UNetBlocks—sometimes incorporating self-attention—process features at the highest resolution. Lower resolutions use UNetBlocks configured for upsampling to restore spatial dimensions incrementally. At each stage, corresponding encoder skip connections are concatenated to fuse fine-grained details. Additional modules such as upsampling convolutions and group normalization further refine outputs.

During forward propagation, noise embeddings are computed first. The encoder extracts hierarchical features while storing skip connections. The decoder then integrates these skips to recover spatial details and enhance reconstruction quality. Auxiliary outputs are finally combined to generate the output image. This design effectively leverages deep hierarchical features and multi-scale fusion, critical for GDD's diffusion-based generation.

## 4 EXPERIMENT

Our experiments validate the effectiveness of GDD in modeling multi-group compositional data across diverse datasets, including CIFAR-10 , CIFAR-100 (Krizhevsky et al., 2009), STL-10[1], and

---

[1] http://ai.stanford.edu/~acoates/stl10

SVHN[2]. To further address dataset complexity, we conducted additional experiments on large-scale, high-variability datasets: CelebA (over 200K diverse human faces) (Liu et al., 2015) and AFHQ (high quality multi category animal facial images) (Choi et al., 2020). GDD generates visually coherent and high-quality facial and animal face images, preserving both fine details (e.g., facial features, hair strands) and global structures such as pose and lighting. In addition, we compare GDD with several established generative frameworks—DDPM (Ho et al., 2020), ViTGAN (Lee et al., 2022), DDIM (Song et al., 2021a), Consistency Models (Song et al., 2023; Salimans & Ho, 2022), Blurring Diffusion (Hoogeboom & Salimans, 2023), AutoGAN (Gong et al., 2019) and Beta diffusion (Zhou et al., 2023), among others—to demonstrate its superiority in generating structured probabilistic representations.

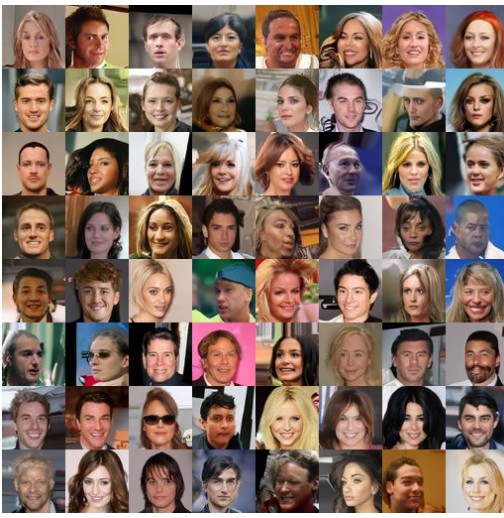 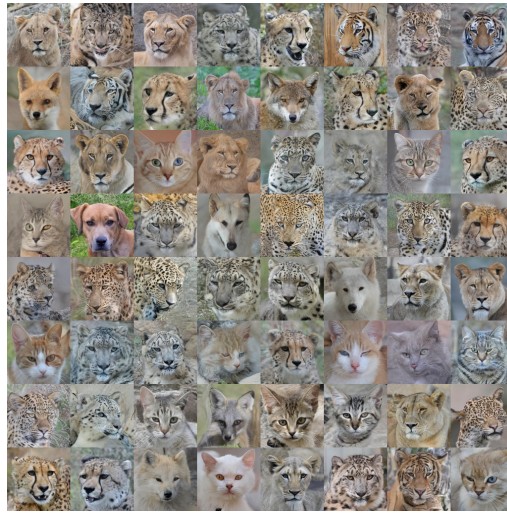

(a) CelebA generated using GDD.        (b) AFHQ generated using GDD.

Figure 3: CelebA (left) and AFHQ (right) images generated using GDD. This figure demonstrates the effectiveness of GDD in generating high-quality images on complex datasets.

**Implementation Details**: We employ the network architecture introduced in Section "Network Architecture" and adopt the scheduling parameter $\alpha_k$ as defined in Equ.(4), thereby enabling more flexible time scheduling adjustments during diffusion model training. The beta diffusion parameters are configured as follows: $S_{\text{shift}} = 0.6$, $S_{\text{scale}} = 0.39$, $c_{\text{start}} = 10$, $c_{\text{end}} = -13$, $\eta = 10000$, batch size $= 256$, $\omega = 0.97$, $\pi = 0.95$. We utilize the Adam optimizer (Kingma & Ba, 2015) with a learning rate of $2 \times 10^{-4}$. For data augmentation, we adopt EDM's approach while restricting the augmented images to the $[0, 1]$ range prior to scaling and shifting. Since GDD extends the beta distribution, we adhere to the original paper's method for limiting the data range. The GDD, trained on 200 million images, is used to compute both the Fréchet Inception Distance (FID) (Heusel et al., 2017) and Kernel Inception Distance (KID) (Binkowski et al., 2018). At the sampling stage, we emulate beta diffusion by generating two types of outputs: one prediction, denoted as $out$, depends solely on the current time step; the other, denoted as $out_1$, leverages the time step parameter $\alpha_{\text{next}}$. Here, $out$ represents a simple single-step prediction, while $out_1$ provides a fine-grained prediction that incorporates a nonlinear sigmoid transformation and time step adjustment. After qualitative observation and quantitative comparison, we chose the former as the actual generated image.

**Comparison Experiment**: To validate the generative capability of GDD, we compare its performance against several state-of-the-art frameworks on the CIFAR-10 dataset. Our experiments are conducted using distributed data parallel training on eight NVIDIA GeForce RTX 4090 GPUs with PyTorch's (Paszke et al., 2019) Distributed DataParallel module, ensuring accelerated optimization and consistent batch processing. We evaluated the quality and diversity of the generated samples using two metrics: FID and KID. FID measures the similarity between the feature distributions of generated and real samples via the Inception-v3 network (Szegedy et al., 2016), while KID com-

---

[2]http://ufldl.stanford.edu/housenumbers

Table 1: Comparison of the FID and KID scores of various generating frameworks trained on CIFAR-10

| Methods | Model | Venue | FID ($\downarrow$) | KID $\times 10^{-3}$ ($\downarrow$) |
|---|---|---|---|---|
| Gaussian Diffusion | DDPM (Ho et al., 2020) | NeurIPS'20 | 3.17 | 2.04 |
| | LSGM (Vahdat et al., 2021) | NeurIPS'21 | 2.10 | - |
| | DDIM (Song et al., 2021a) | ICLR'21 | 15.09 | 10.02 |
| | NCSN++ (Song et al., 2021c) | ICLR'21 | 2.90 | - |
| | Imporved DDPM (Nichol & Dhariwal, 2021) | ICLR'21 | 2.45 | - |
| | GENIE (Dockhorn et al., 2022) | NeurIPS'22 | 5.97 | - |
| | VP-EDM (Karras et al., 2022) | NeurIPS'22 | **1.97** | - |
| | Consistency Models (Song et al., 2023) | ICML'23 | 5.83 | 6.43 |
| | Blackout Diffusion (Santos et al., 2023) | ICML'23 | 4.58 | 2.37 |
| | Blurring Diffusion (Hoogeboom & Salimans, 2023) | ICLR'23 | 3.17 | - |
| GANs | PPOGAN (Wu et al., 2020) | NeurIPS'20 | 10.70 | - |
| | ViTGAN (Lee et al., 2022) | ICLR'22 | 6.66 | - |
| | EAGAN (Ying et al., 2022) | ECCV'22 | 9.91 | - |
| | CLR-GAN (Sun et al., 2024) | ECCV'24 | 23.3 | - |
| | LadaGAN (Morales-Juarez & Pineda, 2024) | arXiv'24 | 3.29 | - |
| Poisson Diffusion | JUMP (Chen & Zhou, 2023) | ICML'23 | 4.80 | - |
| Beta Diffusion | Beta Diffusion (Zhou et al., 2023) | NeurIPS'23 | 3.06 | 1.71 |
| Dirichlet Diffusion | **GDD (Ours)** | - | 2.76 | **1.22** |

putes the maximum mean discrepancy (MMD) between these distributions. Lower values for both metrics indicate better image generation quality.

Table 1 summarizes our findings, showing that GDD demonstrates robust performance across diverse generative models. It outperforms earlier generative models such as DDPM, and GANs in terms of both sample quality and probabilistic fidelity. Compared to recent frameworks like DDIM, Blackout Diffusion and Beta Diffusion, GDD also delivers superior FID and KID scores. Even if the absolute FID is slightly worse than the strongest Gaussian diffusion models(like VP-EDM,NCSN++), GDD remains one of the most effective generative models built on non-Gaussian, simplex-preserving diffusion. The generation of some comparison models in the same experimental setting can be seen in Figure 5.

**Components Ablation**: To validate the effectiveness of our MLP-based mapping network in capturing inter-group dependencies, we conduct ablation studies on the mapping module. Specifically, we examine: (1) **Mapping-removed** (baseline), (2) **Single-layer MLP** (variant), (3) **Full model** (with multi-layer MLP mapping). Quantitative results (Table 2) demonstrate significant performance degradation without mapping. The full multi-layer MLP configuration outperforms both alternatives, confirming that hierarchical feature mixing is essential for modeling complex inter-group interactions, which means the MLP mapping effectively captures inter group dependencies.

Table 2: Ablation study on network components. This table shows the effect of removing or modifying individual components of the proposed network. By comparing different configurations, it highlights each module's contribution to overall performance.

Table 3: Comparison of sampling efficiency between the proposed Grouped Dirichlet Diffusion (GDD) and baseline models, evaluated in terms of peak memory usage (PMU), average processing time per batch (APTPB), and generated images per second (GIPS).

| Network | Parameter | FID ($\downarrow$) | KID $\times 10^{-3}$ ($\downarrow$) |
|---|---|---|---|
| Mapping-removed | 55.47M | 48.14 | 48.21 |
| 1-layer MLP | 55.47M | 47.32 | 45.32 |
| 2-layer MLP | 56M | 6.70 | 5.37 |
| **4-layer MLP (Ours)** | 56.26M | **2.76** | **1.22** |

| Model | PMU | APTPB | GIPS |
|---|---|---|---|
| GDD | 1310.76 MB | 11.25s | 42.8 |
| DDPM | 398.99 MB | 15.27s | 31.5 |
| Beta Diffusion | 1014.54 MB | 13.66s | 37.4 |

**Runtime Complexity:** During training, the GDD framework achieves a stabilized processing rate of 0.69 seconds per kilo-image after initial setup, maintaining remarkable consistency (±0.01 sec/kimg) throughout the training trajectory. Under the same experimental conditions, it represents a 3.3×-7.2× speed advantage over conventional diffusion models like DDPM, which typically require 2-5 seconds per kilo-image. For evaluation, 50,000 images are generated to compute FID and KID scores. The sampling time is approximately 21 minutes at NFE = 200 (equivalent to 42.8 images per second) and 97 minutes at NFE = 1000. Although GDD demonstrates competitive sampling speeds at NFE = 200, surpassing those of traditional diffusion models, there remains potential for further optimization. In addition, Table 3 presents a comparison of peak memory usage, runtime,

and generation speed between GDD and other models, under the same number of sampling steps and generated images.

**Database Comparison And Ablation Study**: Moreover, Figure 3 and Figure 11 illustrates the effect of GDD across different datasets, and Table 4 (a) presents FID scores on these datasets with parameters set to $\eta = 10000$, $B$=256, and $NFE = 1000$. To optimize hyperparameter combinations and further enhance generation results, we conduct experiments on the CIFAR-10 dataset by varying NFE under different combinations of the concentration parameter $\eta$ and mini-batch size $B$. The corresponding FID scores are reported in Table 4 (b). To examine the sensitivity of our method to the KLUB loss weight, we conduct an ablation study over the range ( $\omega \in [0.95, 0.99]$ ). As shown in Table 5, the performance varies noticeably with different choices of ( $\omega$ ). Among all tested configurations, ( $\omega = 0.97$ ) consistently achieves the best FID across different sampling budgets (NFE = 100, 200, 500), indicating that this value provides the most effective balance between the KL guidance and reconstruction terms. Therefore, we adopt ( $\omega = 0.97$ ) as the default setting in all main experiments.

Table 4: (a) Comparison of FID Scores Across Various Datasets. (b) FID scores on CIFAR-10 under varying NFE, $\eta$, and batch size.

| | | | (a) | | | |
|---|---|---|---|---|---|---|
| Dataset | CIFAR-10 | CIFAR-100 | SVHN | STL-10 | CelebA | AFHQ |
| FID | 2.76 | 6.22 | 3.63 | 10.65 | 5.32 | 39.49 |
| | | | (b) | | | |
| $\eta$ | Batchsize | 50 | 100 | 200 | 500 | 1000 |
| 10 | 512 | 42.84 | 34.17 | 30.20 | 25.99 | 25.68 |
| 100 | 512 | 18.48 | 14.35 | 12.47 | 11.45 | 10.40 |
| 1000 | 512 | 9.74 | 6.89 | 5.72 | 4.70 | 4.55 |
| 10000 | 512 | 6.19 | 3.99 | 3.27 | 2.98 | 2.91 |
| 1000 | 256 | 9.73 | 7.07 | 5.90 | 4.90 | 4.82 |
| 10000 | 256 | 6.26 | 3.92 | 3.24 | 2.91 | **2.76** |

## 5 LIMITATION AND FUTURE WORK

Although GDD showed a faster sampling speed compared to traditional diffusion models in this experiment, there is still a lot of room for improvement. Currently, for Gaussian diffusion, various methods have been developed to accelerate the generation of Gaussian diffusion, including combining it with VAEs, GANs, or conditional transport for faster generation , distilling the reverse diffusion chains, utilizing reinforcement learning , and transforming the SDE associated with Gaussian diffusion into an ODE, followed by fast ODE solvers. Given these existing acceleration techniques for Gaussian diffusion, it is worth exploring their generalization to enhance the sampling efficiency of GDD.

## 6 CONCLUSION

We propose GDD, a novel diffusion-based framework designed for hierarchical, multi-group probabilistic data. By replacing the Beta distribution with a Grouped Dirichlet distribution and employing the KLUB loss, GDD enhances computational efficiency, numerical stability, and scalability in high-dimensional settings. Experiments on benchmark datasets demonstrate that GDD outperforms DDPMs, GANs and some advanced diffusion models, effectively modeling hierarchical structures through independent Dirichlet groups and logit-space operations. Future work will explore extensions to text-to-image synthesis and large-scale language models. This study establishes a crucial link between diffusion models and structured probabilistic data, providing a robust, scalable framework for advancing generative modeling.

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

# A  APPENDIX

## A.1  ADDITIONAL EXPERIMENTS

Table 5: Ablation study on weight parameters $\omega$, FID scores on CIFAR-10 under varying NFE.

| Weight parameters $\omega$ | Batchsize | NFE=100 | NFE=200 | NFE=500 |
|:---:|:---:|:---:|:---:|:---:|
| $\omega$=0.95 | 256 | 4.58 | 3.29 | 3.21 |
| $\omega$=0.96 | 256 | 4.51 | 3.81 | 3.04 |
| $\omega$=0.97 | 256 | 3.92 | 3.24 | 2.91 |
| $\omega$=0.98 | 256 | 5.51 | 4.75 | 3.80 |
| $\omega$=0.99 | 256 | 5.12 | 4.74 | 3.93 |

Table 6: FID scores on CIFAR-10 under different grouping strategies (same batch size), evaluated across multiple NFEs.

| Grouping Strategies | NFE=100 | NFE=200 | NFE=500 | NFE=1000 |
|:---:|:---:|:---:|:---:|:---:|
| Color channels partitioning | 3.92 | 3.24 | 2.91 | 2.76 |
| Spatial pixel patching(Regular Division) | 4.18 | 3.53 | 3.01 | 2.94 |
| Spatial pixel patching(Irregular Division) | 328 | 298 | 288 | 273 |
| Random feature partitioning | 522 | 410 | 403 | 401 |

**Grouped Experiment**: In addition to grouping channels into semantically meaningful RGB feature sets, we further evaluated the generality of our grouped Dirichlet diffusion framework using two alternative grouping strategies: spatial pixel patching and random feature partitioning. Experiments on CIFAR-10 (both qualitative samples and quantitative FID scores) show that spatial pixel patching yields reasonable performance when the image is divided into only a few coarse blocks by rules. However, its results remain slightly inferior to RGB grouping, as spatial partitioning disrupts the natural coherence of color distributions across the image. When the number of spatial patches increases, training and sampling become substantially slower while FID exhibits no meaningful improvement. When the division method becomes irregular, quantitative indicators will become worse. In contrast, random feature partitioning severely breaks semantic structure: generated samples become visually incoherent, and quantitative metrics deteriorate sharply. This confirms that the model struggles to learn stable dependency patterns under arbitrary, non-semantic groupings. Figure 4 illustrates the two grouping strategies—RGB grouping and a four-patch spatial partition—while Table 6 reports the corresponding FID scores under varying NFEs. The quantitative results consistently show that

RGB grouping remains the most effective, highlighting the importance of semantically meaningful grouping for GDD. The generation effects of these grouping strategies can be seen in Figure 6.

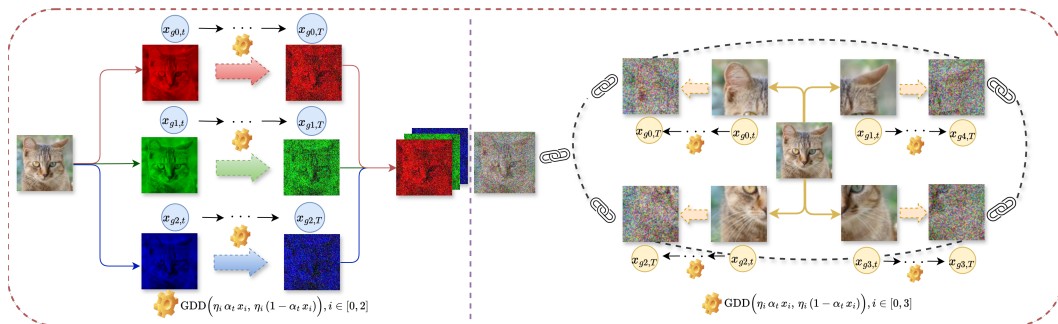

Figure 4: **Illustration of two grouping strategies in Grouped Dirichlet Diffusion**: The left panel shows grouping by RGB color channels, while the right panel shows grouping via spatially partitioned pixel regions. Both approaches preserve intra-group dependencies and allow the diffusion process to dynamically adapt inter-group interactions over time.

## A.2 DEFINITION

**Definition 1 (Grouped Dirichlet Distribution(Ng et al., 2008))** *Let $G$ denote the number of independent groups. For each group $g \in \{1, \ldots, G\}$, $\mathbf{x}_g = (x_{g1}, x_{g2}, \ldots, x_{gK})$ is the random vector, it lies on the $K$-dimensional simplex, meaning it satisfies:*

$$\sum_{i=1}^{K} x_{gi} = 1, \quad x_{gi} \geq 0 \quad \text{for all } i \in \{1, \ldots, K\}. \tag{20}$$

*If each group $\mathbf{x}_g$ independently follows a Dirichlet distribution with parameter vector $\boldsymbol{\alpha}_g = (\alpha_{g1}, \ldots, \alpha_{gK})$, then the joint distribution of all groups is referred to as the grouped dirichlet Distribution. Formally, it is defined as:*

$$p\left(\{\mathbf{x}_g\}_{g=1}^G; \{\boldsymbol{\alpha}_g\}_{g=1}^G\right) = \prod_{g=1}^{G} Dir\left(\mathbf{x}_g; \boldsymbol{\alpha}_g\right). \tag{21}$$

*The probability density function (PDF) for a single group's Dirichlet distribution is given by:*

$$Dir\left(\mathbf{x}_g; \boldsymbol{\alpha}_g\right) = \frac{1}{B(\boldsymbol{\alpha}_g)} \prod_{i=1}^{K} x_{gi}^{\alpha_{gi}-1}, \tag{22}$$

*where $B(\boldsymbol{\alpha}_g)$ is the multivariate beta function that serves as the normalization constant:*

$$B(\boldsymbol{\alpha}_g) = \frac{\prod_{i=1}^{K} \Gamma(\alpha_{gi})}{\Gamma\left(\sum_{i=1}^{K} \alpha_{gi}\right)}, \tag{23}$$

*where, $\Gamma(\cdot)$ denotes the gamma function.*

## A.3 LOSS FUNCTION DESIGN

Since the Grouped Dirichlet distribution extends the Beta distribution, we adopt the KL Upper Bound (KLUB) originally proposed for Beta Diffusion (Zhou et al., 2023)—whose feasibility has been well established—and discuss its application to the multigroup Dirichlet distribution.

For a single group, consider two Dirichlet distributions defined over a $K$-dimensional simplex, $\text{Dir}(x; \boldsymbol{\alpha})$ and $\text{Dir}(x; \boldsymbol{\beta})$, where $\boldsymbol{\alpha} = (\alpha_1, \alpha_2, \ldots, \alpha_K)$, $\boldsymbol{\beta} = (\beta_1, \beta_2, \ldots, \beta_K)$. Let $\alpha_0 = \sum_{i=1}^{K} \alpha_i$, $\beta_0 = \sum_{i=1}^{K} \beta_i$. The KL divergence from $\text{Dir}(x; \boldsymbol{\alpha})$ to $\text{Dir}(x; \boldsymbol{\beta})$ is then given by: Let

Table 7: Summary of Key Notations. This table outlines the primary symbols and their definitions used throughout the paper for clarity and consistency.

| Symbol | Meaning |
|---|---|
| $K$ | Components per group (simplex dimension $K-1$) |
| $g$ | Group index |
| $\mathbf{x}_{g0}$ | Clean data for group $g$ ($\in \Delta_{K-1}$) |
| $\mathbf{Z}_g(t), \mathbf{z}_g^{(k)}$ | State at time $t$ / step $k$ |
| $\alpha(t), \alpha_k$ | Sigmoid schedule (continuous / discrete) |
| $\lambda(t), \lambda_k$ | Signal-to-noise decay rate |
| $\eta$ | Dirichlet concentration (global noise level) |
| $\Sigma(\mathbf{z})$ | Wright–Fisher covariance $\mathrm{diag}(\mathbf{z}) - \mathbf{z}\mathbf{z}^\top$ |
| $B(\mathbf{z}), B_k$ | Any matrix with $BB^\top = 2\Sigma(\mathbf{z})$ |
| $\mathbf{W}_g(t), \bar{\mathbf{W}}_g(t)$ | $K$-dim. Brownian motions (forward / reverse) |
| $h$ | Time-step size $1/T$ |
| $f_\theta(\cdot)$ | Generator predicting the clean signal |
| $s_\theta(\mathbf{z}, t)$ | Learned score $\nabla_{\mathbf{z}} \log q_t(\mathbf{z})$ |
| $\boldsymbol{\varepsilon}_{g,k}, \bar{\boldsymbol{\varepsilon}}_{g,k}$ | i.i.d. $\mathcal{N}(\mathbf{0}, \mathbf{I}_K)$ |

$\boldsymbol{\alpha} = (\alpha_1, \ldots, \alpha_K), \qquad \boldsymbol{\beta} = (\beta_1, \ldots, \beta_K)$, and define $\alpha_0 = \sum_{i=1}^K \alpha_i, \qquad \beta_0 = \sum_{i=1}^K \beta_i$. The KL divergence between two Dirichlet distributions is defined as:

$$\mathrm{KL}\Big(\mathrm{Dir}(\boldsymbol{\alpha}) \;\|\; \mathrm{Dir}(\boldsymbol{\beta})\Big) = \int_{\Delta_{K-1}} p_{\boldsymbol{\alpha}}(\mathbf{x}) \ln \frac{p_{\boldsymbol{\alpha}}(\mathbf{x})}{p_{\boldsymbol{\beta}}(\mathbf{x})} \, \mathrm{d}\mathbf{x}, \tag{24}$$

where the simplex is defined as $\Delta_{K-1} = \left\{ \mathbf{x} \in \mathbb{R}^K_{\geq 0} \,\middle|\, \sum_{i=1}^K x_i = 1 \right\}$. For the Dirichlet densities:

$$p_{\boldsymbol{\alpha}}(\mathbf{x}) = \frac{1}{B(\boldsymbol{\alpha})} \prod_{i=1}^K x_i^{\alpha_i - 1}, \quad p_{\boldsymbol{\beta}}(\mathbf{x}) = \frac{1}{B(\boldsymbol{\beta})} \prod_{i=1}^K x_i^{\beta_i - 1}, \tag{25}$$

the log–density ratio is given by:

$$\ln \frac{p_{\boldsymbol{\alpha}}(\mathbf{x})}{p_{\boldsymbol{\beta}}(\mathbf{x})} = \ln \frac{B(\boldsymbol{\beta})}{B(\boldsymbol{\alpha})} + \sum_{i=1}^K (\alpha_i - \beta_i) \ln x_i. \tag{26}$$

Using the multivariate beta function $B(\boldsymbol{\alpha}) = \frac{\prod_{i=1}^K \Gamma(\alpha_i)}{\Gamma(\alpha_0)}$, we have the following:

$$\ln \frac{B(\boldsymbol{\beta})}{B(\boldsymbol{\alpha})} = \ln \frac{\Gamma(\alpha_0)}{\Gamma(\beta_0)} - \sum_{i=1}^K \ln \frac{\Gamma(\alpha_i)}{\Gamma(\beta_i)}. \tag{27}$$

A key identity for the Dirichlet distribution is:

$$\mathbb{E}_{\mathbf{x} \sim \mathrm{Dir}(\boldsymbol{\alpha})}\big[\ln x_i\big] = \psi(\alpha_i) - \psi(\alpha_0), \tag{28}$$

where $\psi(\cdot) = \frac{\mathrm{d}}{\mathrm{d}z} \ln \Gamma(z)$ is the digamma function. Substituting equation 27–equation 28 into Eq. equation 24 yields the following:

$$\begin{aligned} \mathrm{KL}(\boldsymbol{\alpha} \;\|\; \boldsymbol{\beta}) = {} & \ln \frac{\Gamma(\alpha_0)}{\Gamma(\beta_0)} - \sum_{i=1}^K \ln \frac{\Gamma(\alpha_i)}{\Gamma(\beta_i)} \\ & + \sum_{i=1}^K (\alpha_i - \beta_i)\left[\psi(\alpha_i) - \psi(\alpha_0)\right]. \end{aligned} \tag{29}$$

For the Grouped Dirichlet distribution, define:

$$p = \prod_{g=1}^{G} \mathrm{Dir}(\boldsymbol{\alpha}_g), \quad q = \prod_{g=1}^{G} \mathrm{Dir}(\boldsymbol{\beta}_g). \tag{30}$$

The KL divergence between these two distributions is then:

$$D_{\mathrm{KL}}(p\|q) = \sum_{g=1}^{G} D_{\mathrm{KL}}\Big(\mathrm{Dir}(\boldsymbol{\alpha}_g) \,\|\, \mathrm{Dir}(\boldsymbol{\beta}_g)\Big). \tag{31}$$

We refer to this sum—equivalent to the Bregman divergence associated with the log-beta function—as the log-beta divergence.

# B  THEORETICAL ANALYSIS

Let $G$ be the number of independent groups, each of dimension $K$. For every group $g \in \{1, \dots, G\}$ we define $\mathbf{a}_g = (\alpha_{g1}, \dots, \alpha_{gK})$, $\alpha_{0g} = \sum_{i=1}^{K} \alpha_{gi}$, and fix a global concentration parameter $\eta > 0$ together with a monotonically decreasing noise schedule $\alpha_t \in (0, 1]$ for $t \in [0, 1]$. The clean data are denoted by $\mathbf{x}_0 = \{\mathbf{x}_{g0}\}_{g=1}^{G}$, with $\mathbf{x}_{g0} \sim \mathrm{Dir}(\eta\,\mathbf{a}_g)$. All random variables reside on the $(K-1)$-simplex $\Delta_{K-1} := \{\mathbf{x} \in \mathbb{R}_{\geq 0}^{K} \mid \sum_{i=1}^{K} x_i = 1\}$.

**Theorem 1 (Forward Closure)** *For any $t \in [0, 1]$, the forward diffusion marginal $q(\mathbf{z}_t \mid \mathbf{x}_0) = \prod_{g=1}^{G} Dir\Big(\mathbf{z}_{g,t}; \eta\alpha_t\mathbf{a}_g\Big)$ remains in the* Grouped Dirichlet *family. Moreover, $\mathbb{E}[\mathbf{z}_{g,t} \mid \mathbf{x}_{g0}] = \alpha_t\,\mathbf{x}_{g0}$.*

***Proof** 1 Fix a group $g$. Draw independent Gamma variables $y_{gi} \sim \mathrm{Gamma}\Big(\eta\alpha_t\alpha_{gi}, 1\Big)$, for $i = 1, \dots, K$. Define $S_g = \sum_{i=1}^{K} y_{gi}$, and $\mathbf{z}_{g,t} = \frac{1}{S_g}(y_{g1}, \dots, y_{gK})$. By the standard Gamma–Dirichlet equivalence, it follows that $\mathbf{z}_{g,t} \sim Dir(\eta\alpha_t\mathbf{a}_g)$.*

*Independence across groups yields the product form for $q(\mathbf{z}_t \mid \mathbf{x}_0)$. For the expectation, note that the ith component of a Dirichlet vector satisfies $\mathbb{E}[z_{g,t}^{(i)}] = \frac{\eta\alpha_t\alpha_{gi}}{\eta\alpha_t\alpha_{0g}} = \alpha_t x_{g0}^{(i)}$. Thus, $\mathbb{E}[\mathbf{z}_{g,t} \mid \mathbf{x}_{g0}] = \alpha_t\,\mathbf{x}_{g0}$.*

**Theorem 2 (Time–Separable Conditional)** *For $0 \leq s < t \leq 1$, $q(\mathbf{z}_s \mid \mathbf{z}_t, \mathbf{x}_0) = \prod_{g=1}^{G} Dir\Big(\mathbf{z}_{g,s}; \eta(\alpha_s - \alpha_t)\mathbf{a}_g\Big)$.*

***Proof** 2 Fix a group $g$. Represent the Gamma variable at time $t$ as a sum: $y_{gi}^{(t)} = y_{gi}^{(s)} + y_{gi}^{(\Delta)}$, where $y_{gi}^{(s)} \sim \mathrm{Gamma}\Big(\eta\alpha_s\alpha_{gi}, 1\Big)$, and $y_{gi}^{(\Delta)} \sim \mathrm{Gamma}\Big(\eta(\alpha_t - \alpha_s)\alpha_{gi}, 1\Big)$ with all variables mutually independent. Conditioning on $\mathbf{z}_{g,t}$ is equivalent to conditioning on the ratios $r_{gi} = \frac{y_{gi}^{(t)}}{S_g^{(t)}}$, with $S_g^{(t)} = \sum_{i=1}^{K} y_{gi}^{(t)}$. The Gamma Partition Theorem (Kotz et al., 2019) implies that, given the total $S_g^{(t)}$, the vector $(y_{g1}^{(s)}, \dots, y_{gK}^{(s)})$ follows a Dirichlet distribution with parameters $\eta\alpha_s\mathbf{a}_g$. After normalizing by its own sum, we obtain $\mathbf{z}_{g,s} \mid \mathbf{z}_{g,t} \sim Dir(\eta\alpha_s\mathbf{a}_g)$. Subtracting the common part yields the desired increment, $Dir\Big(\eta(\alpha_s - \alpha_t)\mathbf{a}_g\Big)$. The independence across groups then gives the stated product expression.*

**Theorem 3 (Convergence of the Reverse Chain)** *Define*

$$\varepsilon(\theta) := \sup_{t \in [0,1]} \sum_{g=1}^{G} \Big\| \hat{\boldsymbol{\alpha}}_{g,\theta}(t) - \eta\alpha_t\mathbf{a}_g \Big\|_1 \xrightarrow[\theta \to \theta^\star]{} 0. \tag{32}$$

*Let $\mu_{0,\theta}$ denote the distribution obtained by running the learned reverse Markov chain $p_\theta(\mathbf{z}_{k-1} \mid \mathbf{z}_k)$ for $T$ steps, starting from an arbitrary prior at $t = 1$. Then, there exists a constant $C = C(T, \eta, \alpha_{\min})$ such that*

$$\|\mu_{0,\theta} - \mathcal{D}_{\mathbf{x}_0}\|_{TV} \leq C\,\varepsilon(\theta), \tag{33}$$

*where $\mathcal{D}_{\mathbf{x}_0}$ is the true data distribution. Consequently, $\mu_{0,\theta} \xrightarrow{TV} \mathcal{D}_{\mathbf{x}_0}$ as $\varepsilon(\theta) \to 0$.*

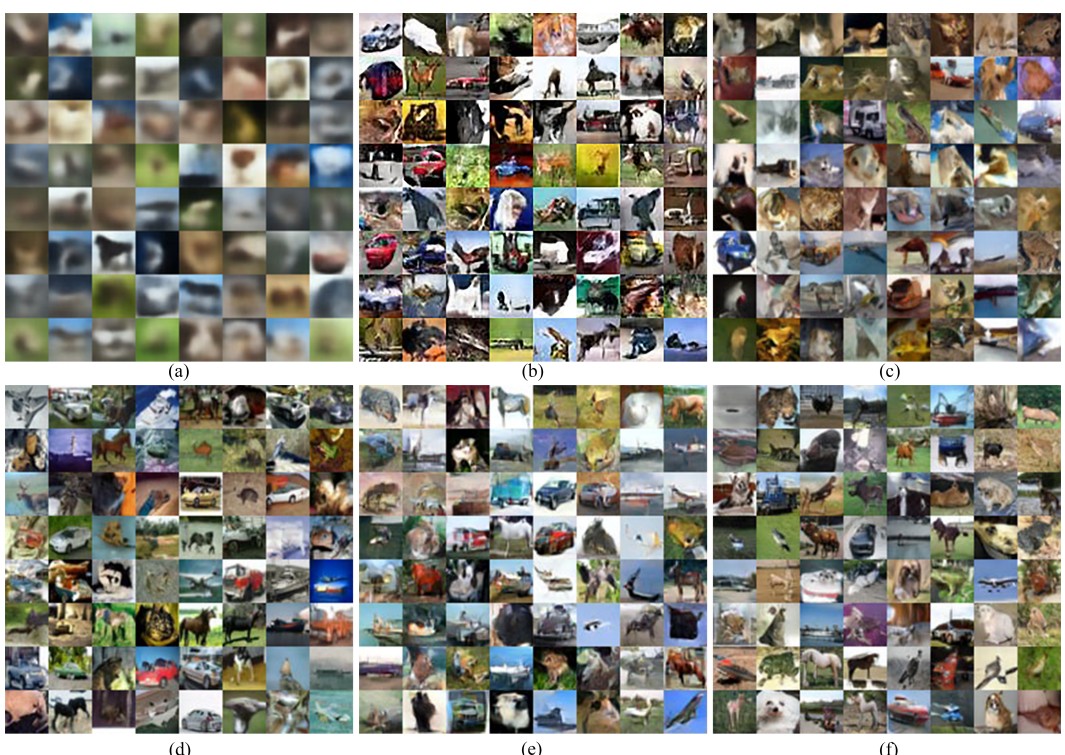

Figure 5: (a) CIFAR-10 images generated using VAE; (b) CIFAR-10 images generated using GAN; (c) CIFAR-10 images generated using DDPM; (d) CIFAR-10 images generated using DDIM; (e) CIFAR-10 images generated using Consistency Models; and (f) CIFAR-10 images generated using AutoGAN.

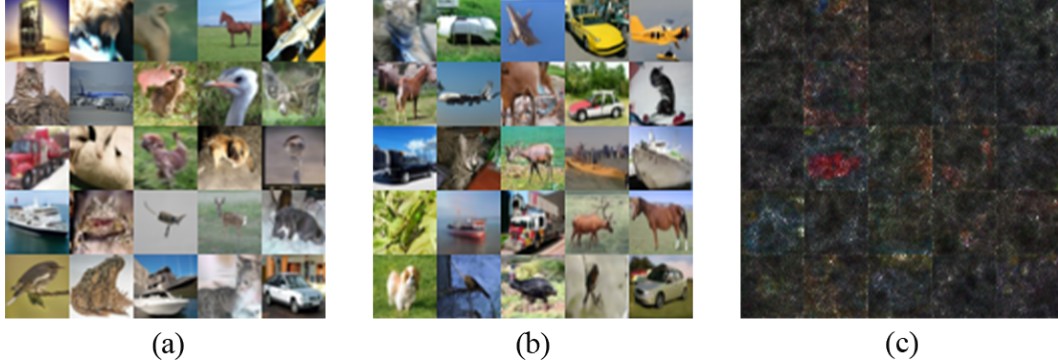

Figure 6: (a) CIFAR-10 images generated using color-channel partitioning; (b) CIFAR-10 images generated using four-patch spatial partitioning; (c) CIFAR-10 images generated using random feature partitioning.

**Theorem 4 (Monotone Entropy)** *Let $H_g(t)$ denote the differential entropy of group $g$ at time $t$. Then*

$$\frac{d}{dt} H_g(t) = -\eta \, \dot{\alpha}_t \, \psi_1\Big(\eta \alpha_t \alpha_{0g}\Big) \alpha_{0g} \; < \; 0, \tag{34}$$

*where $\psi_1$ is the trigamma function. Hence, the entropy increases strictly along the forward diffusion.*

***Proof 3*** *Let $\mathbf{Z} \sim Dir(\boldsymbol{\beta})$ with $\beta_0 = \sum_i \beta_i$. Its differential entropy is given by $H(\mathbf{Z}) = \ln B(\boldsymbol{\beta}) + (\beta_0 - K)\psi(\beta_0) - \sum_{i=1}^{K}(\beta_i - 1)\psi(\beta_i)$, where $\psi$ denotes the digamma function. Substituting $\beta_i = \eta \alpha_t \alpha_{gi}$ and differentiating with respect to $t$ (using $\frac{d}{dt}\beta_i = \eta \dot{\alpha}_t \alpha_{gi}$ and $\psi_1 = \frac{d}{dz}\psi$), all terms cancel except for $-\eta \, \dot{\alpha}_t \, \alpha_{0g} \, \psi_1\Big(\eta \alpha_t \alpha_{0g}\Big)$. Since $\dot{\alpha}_t < 0$ and $\psi_1 > 0$, the derivative is negative.*

**Theorem 5 (Consistency of KL Upper Bound)** *Define the training objective as*

$$\mathcal{L}_{\mathrm{KLUB}}(\theta) = \sum_{g=1}^{G} \mathrm{KL}\Big(\mathrm{Dir}(\eta \alpha_s \mathbf{a}_g) \,\big\|\, \mathrm{Dir}(\hat{\boldsymbol{\alpha}}_{g,\theta})\Big) \tag{35}$$
$$\hat{\boldsymbol{\alpha}}_{g,\theta} := f_\theta(\mathbf{z}_t, t).$$

*If a parameter vector $\theta^\star$ satisfies $\hat{\boldsymbol{\alpha}}_{g,\theta^\star} = \eta \alpha_s \mathbf{a}_g$, for every g, then $\mathcal{L}_{\mathrm{KLUB}}(\theta) \geq 0$, $\quad \mathcal{L}_{\mathrm{KLUB}}(\theta^\star) = 0$, and $\theta^\star$ simultaneously minimizes the negative log-likelihood $-\log p_\theta(\mathbf{z}_s \mid \mathbf{z}_t)$.*

***Proof 4*** *By Gibbs' inequality (Kvalseth, 1997), $KL(P\|Q) \geq 0$ with equality if and only if $P = Q$ almost everywhere. Therefore, $\mathcal{L}_{\mathrm{KLUB}}(\theta) \geq 0$ and equals zero precisely when $\theta = \theta^\star$. For a fixed $\mathbf{z}_s$, the negative log-likelihood is given by*

$$-\log p_\theta(\mathbf{z}_s \mid \mathbf{z}_t) = \sum_{g=1}^{G}\Big[\ln B\Big(\hat{\boldsymbol{\alpha}}_{g,\theta}\Big) - (\hat{\boldsymbol{\alpha}}_{g,\theta} - \mathbf{1}) \cdot \ln \mathbf{z}_{g,s}\Big],$$

*which differs from $\mathcal{L}_{\mathrm{KLUB}}(\theta)$ only by constants independent of $\theta$. Hence, both objectives share the same minimizer $\theta^\star$.*

**Theorem 6 (Convergence of the Reverse Chain)** *Define*

$$\varepsilon(\theta) := \sup_{t \in [0,1]} \sum_{g=1}^{G}\Big\|\hat{\boldsymbol{\alpha}}_{g,\theta}(t) - \eta \alpha_t \mathbf{a}_g\Big\|_1 \xrightarrow[\theta \to \theta^\star]{} 0. \tag{36}$$

*Let $\mu_{0,\theta}$ denote the distribution obtained by running the learned reverse Markov chain $p_\theta(\mathbf{z}_{k-1} \mid \mathbf{z}_k)$ for $T$ steps, starting from an arbitrary prior at $t = 1$. Then, there exists a constant $C = C(T, \eta, \alpha_{\min})$ such that*

$$\|\mu_{0,\theta} - \mathcal{D}_{\mathbf{x}_0}\|_{TV} \leq C \, \varepsilon(\theta), \tag{37}$$

*where $\mathcal{D}_{\mathbf{x}_0}$ is the true data distribution. Consequently, $\mu_{0,\theta} \xrightarrow{TV} \mathcal{D}_{\mathbf{x}_0}$ as $\varepsilon(\theta) \to 0$.*

***Proof 5*** *Step 1 (Lipschitz Property). For two Dirichlet densities, $f_{Dir(\boldsymbol{\beta})}$ and $f_{Dir(\boldsymbol{\gamma})}$, with parameters bounded below by a positive constant, Scheffé's lemma combined with mean-value estimates on $\partial_{\boldsymbol{\beta}} f$ yields $\|Dir(\boldsymbol{\beta}) - Dir(\boldsymbol{\gamma})\|_{TV} \leq L \|\boldsymbol{\beta} - \boldsymbol{\gamma}\|_1$, for some finite constant $L$ independent of $\boldsymbol{\beta}$ and $\boldsymbol{\gamma}$.*

***Step 2 (One-Step Error Propagation).*** *Let $\nu_t$ and $\pi_t$ denote the model and true distributions at time $t$, respectively. For the reverse kernel $K_\theta^{(t \to s)}$, we have $\|\nu_s - \pi_s\|_{TV} \leq \|\nu_t - \pi_t\|_{TV} + \sup_{\mathbf{z}_t}\Big\|K_\theta^{(t \to s)}(\mathbf{z}_t, \cdot) - K_{\theta^\star}^{(t \to s)}(\mathbf{z}_t, \cdot)\Big\|_{TV}$. By Step 1, the second term is bounded by $L \varepsilon(\theta)$. Dividing the interval $[0,1]$ into $T$ equal steps and iterating the above inequality yields $\|\mu_{0,\theta} - \mathcal{D}_{\mathbf{x}_0}\|_{TV} \leq T L \varepsilon(\theta) = C \varepsilon(\theta)$. Thus, if $\varepsilon(\theta) \to 0$, then $\mu_{0,\theta} \to \mathcal{D}_{\mathbf{x}_0}$ in total variation.*

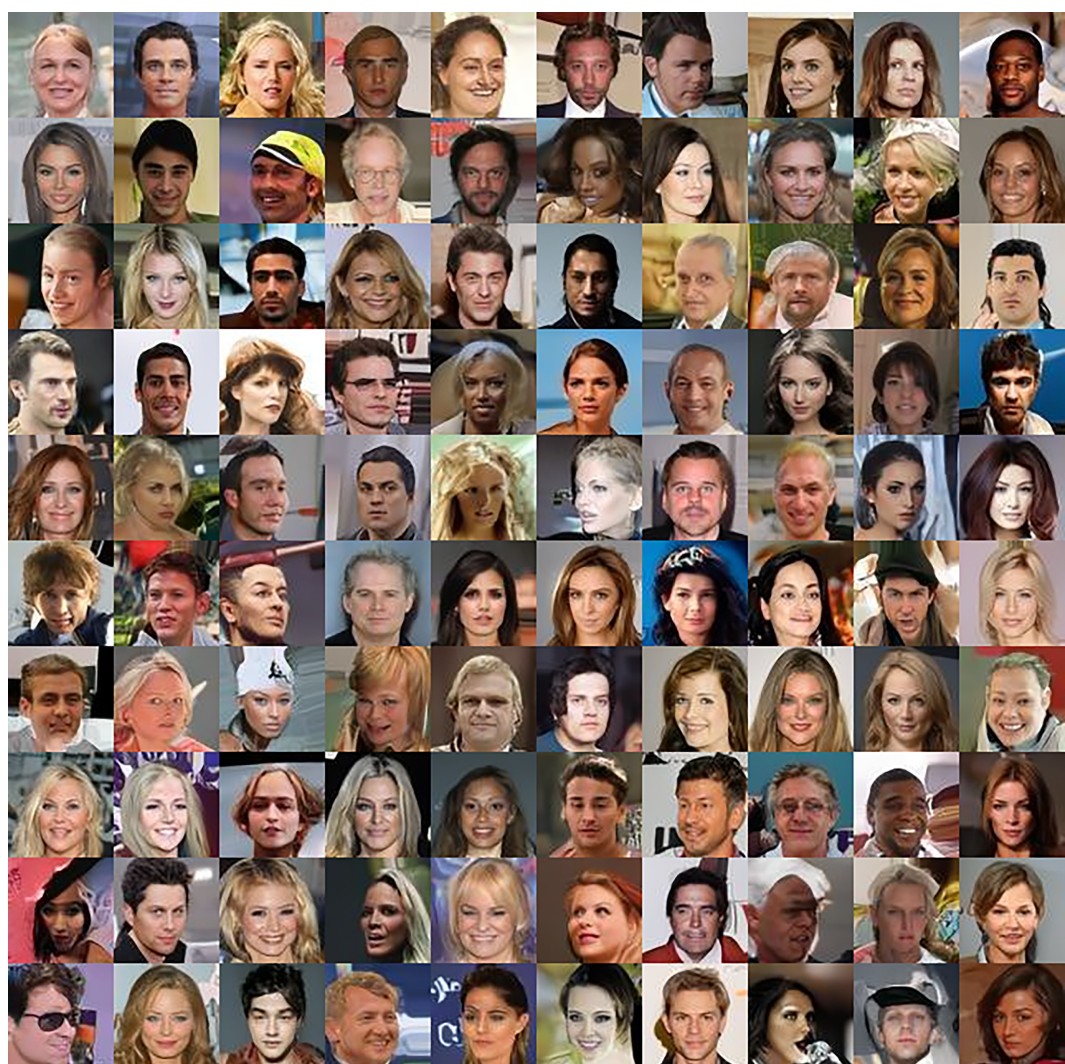

Figure 7: Representative $64 \times 64$ face images generated by our model on the CelebA dataset, illustrating both high visual fidelity and a wide diversity of facial attributes. This figure demonstrates that the proposed model is capable of generating realistic and visually appealing face images at $64 \times 64$ resolution. The images exhibit a high degree of visual fidelity, accurately capturing fine facial details. Additionally, the generated faces display a broad range of facial attributes (such as age, gender, hairstyle, and expression), highlighting the model's ability to produce diverse and high-quality samples that reflect the complexity of the CelebA dataset.

### B.1 CONTINUOUS–TIME SDE FORMULATION OF GDD

**Forward process (data → noise).** Building on the framework introduced in Score-Based Generative Modeling through Stochastic Differential Equations (Song et al., 2021b), we derive the following results. For each group $g$, the state vector $\mathbf{Z}_g(t) \in \Delta_{K-1}$ evolves on the probability simplex as:

$$d\mathbf{Z}_g(t) = \lambda(t)\big(\mathbf{x}_{g0} - \mathbf{Z}_g(t)\big)\, dt + \sqrt{\frac{\lambda(t)}{\eta}}\, B\big(\mathbf{Z}_g(t)\big)\, d\mathbf{W}_g(t), \tag{38}$$

where $\lambda(t) = -\dot{\alpha}(t)/\alpha(t) \geq 0$ is induced by the *sigmoid schedule* $\alpha(t) \in (0,1]$. **Reverse process (noise → data).** Given the learned score $s_\theta(\mathbf{z}, t) = \nabla_\mathbf{z} \log q_t(\mathbf{z})$ and network prediction $\hat{\mathbf{x}}_{g,\theta}(\mathbf{z}, t)$,

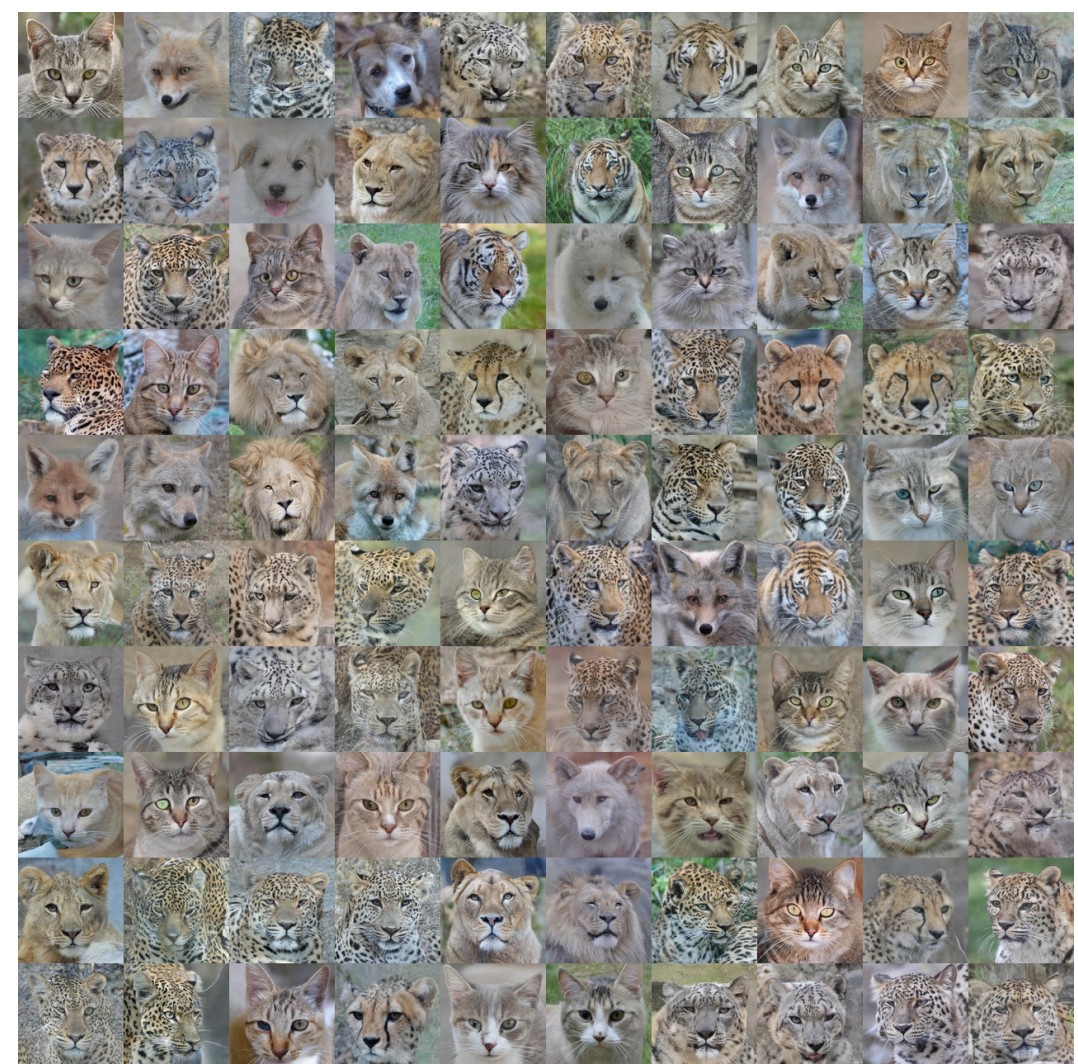

Figure 8: Qualitative results of the proposed Grouped Dirichlet Diffusion (GDD) model on the AFHQ dataset. The figure presents $128 \times 128$ images synthesized by GDD, demonstrating high-fidelity textures, clear species characteristics, and notable diversity across samples.

the reverse-time SDE is:

$$
\mathrm{d}\mathbf{Z}_g(t) = \Big[\lambda(t)\big(\widehat{\mathbf{x}}_{g,\theta}(\mathbf{Z}_g, t) - \mathbf{Z}_g(t)\big) - \frac{\lambda(t)}{\eta}\,\Sigma\big(\mathbf{Z}_g(t)\big)s_\theta(\mathbf{Z}_g, t)\Big]\mathrm{d}t
$$
$$
+ \sqrt{\frac{\lambda(t)}{\eta}}\,B\big(\mathbf{Z}_g(t)\big)\,\mathrm{d}\bar{\mathbf{W}}_g(t),
\tag{39}
$$

$$
\mathrm{d}\mathbf{Z}_g(t) = \Big[\lambda(t)\big(\widehat{\mathbf{x}}_{g,\theta}(\mathbf{Z}_g, t) - \mathbf{Z}_g(t)\big) - \frac{\lambda(t)}{\eta}\,\Sigma\big(\mathbf{Z}_g(t)\big)s_\theta(\mathbf{Z}_g, t)\Big]\mathrm{d}t
$$
$$
+ \sqrt{\frac{\lambda(t)}{\eta}}\,B\big(\mathbf{Z}_g(t)\big)\,\mathrm{d}\bar{\mathbf{W}}_g(t),
\tag{40}
$$

where $BB^\top = 2\Sigma$ with $\Sigma(\mathbf{z}) = \mathrm{diag}(\mathbf{z}) - \mathbf{z}\mathbf{z}^\top$.

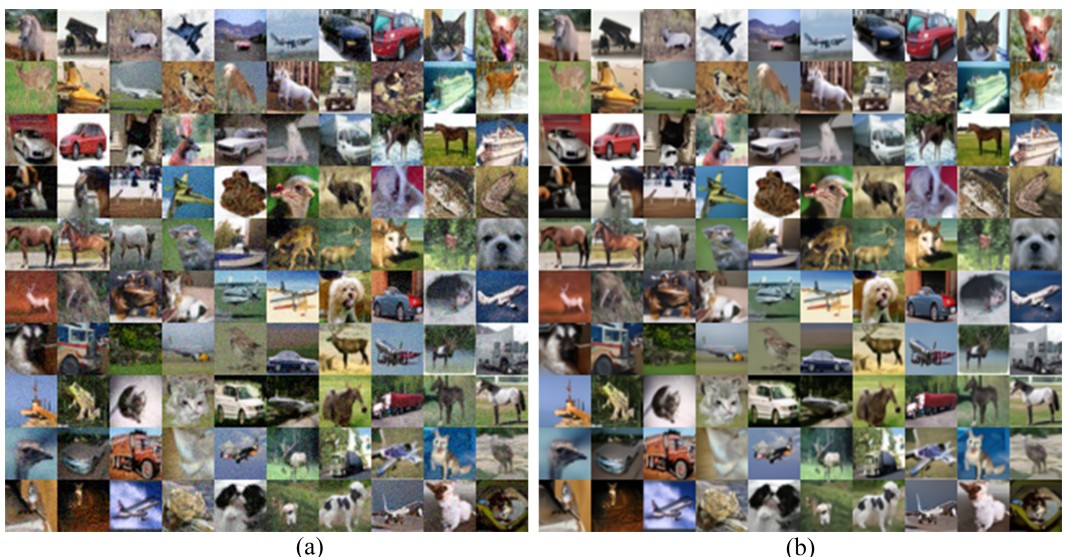

(a)                                        (b)

Figure 9: The figure presents two outputs from the GDD model on CIFAR-10. In panel (a), the output $out_1$ is determined by the time step parameter $\alpha_{next}$, which incorporates a nonlinear transformation. In contrast, panel (b) shows an output $out$ that depends solely on the current time step, representing a straightforward single-step prediction.

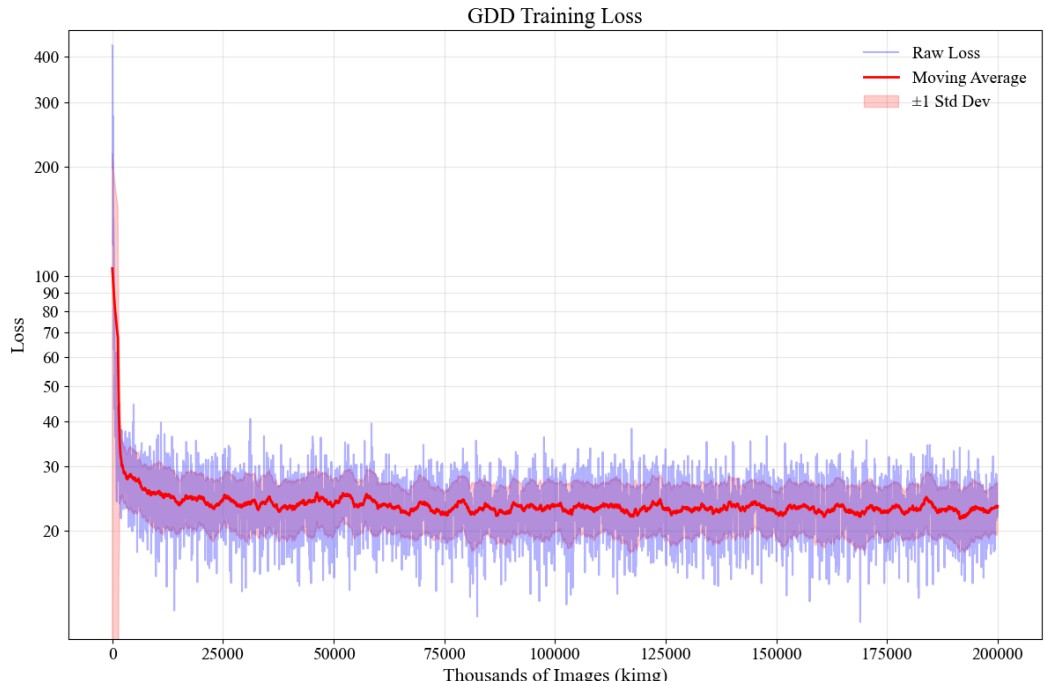

Figure 10: Training loss curve of the proposed GDD model. The horizontal axis shows the number of thousands of images (kimg) processed during training, and the vertical axis shows the loss in logarithmic scale.

---

**Algorithm 1:** Training of Grouped Dirichlet Diffusion (GDD)

---

    // **\*Training\*** **Input:** Dataset $\mathcal{D}$ (each sample contains groups $G$ with $\{\mathbf{x}_g\}$); mini-batch size $\mathcal{B}$; concentration parameter $\eta$; data shifting $S_{\text{shift}}$; data scaling $S_{\text{scale}}$; generator $f_\theta$; loss balance coefficient $\omega$; time reversal coefficient $\pi$; scheduling function $\alpha_t$ (e.g., beta linear or sigmoid schedule).

**Initialization:** Initialize the parameters of the generator network $f_\theta$.

**while** *not converged* **do**

    Draw mini-batch $X_0 = \{x_0^{(i)}\}_{i=1}^{\mathcal{B}}$ from $\mathcal{D}$;

    **for** $i = 1, 2, \ldots, \mathcal{B}$ **do**

        Sample $t_i \sim \text{Unif}(0, 1)$;

        Compute $s_i = \pi\, t_i$;

        Compute scheduling parameters $\alpha_{t_i}$ and $\alpha_{s_i}$ via $\alpha_t$;

        Scale and shift input: $x_0^{(i)} \leftarrow x_0^{(i)} \times S_{\text{scale}} + S_{\text{shift}}$;

        **for** *each group $g$ in sample $i$* **do**

            Generate $\mathbf{z}_{g,t_i}^{(i)} \sim \text{Dir}\left(\eta\, \alpha_{t_i}\, \mathbf{x}_{g,0}^{(i)}\right)$;

        **end**

        Compute prediction: $\hat{x}_0^{(i)} = f_\theta\left(\mathbf{z}_{t_i}^{(i)}, t_i\right) \times S_{\text{scale}} + S_{\text{shift}}$;

        Compute loss:

$$\mathcal{L}_i = \omega\, \text{KLUB}\left(s_i, \mathbf{z}_{t_i}^{(i)}, x_0^{(i)}\right) + (1 - \omega)\, \text{KLUB}\left(\mathbf{z}_{t_i}^{(i)}, x_0^{(i)}\right)$$

        // Replace beta-based KL with Grouped Dirichlet-based KL; use each group $g$'s Dirichlet parameters for computation.

    **end**

    Update $f_\theta$ by performing SGD with

$$\frac{1}{\mathcal{B}} \nabla_\theta \sum_{i=1}^{\mathcal{B}} \mathcal{L}_i.$$

**end**

                  // **Output:** Trained network parameters $f_\theta$.

---

### B.2 Discrete–Time SDE Formulation

**Forward Euler–Maruyama Step** Let $t_k = k/T$ and $h = 1/T$,

$$\mathbf{z}_g^{(k+1)} = \mathbf{z}_g^{(k)} + h\, \lambda_k\left(\mathbf{x}_{g0} - \mathbf{z}_g^{(k)}\right)$$

$$+ \sqrt{\frac{h\lambda_k}{\eta}}\, B_k\, \boldsymbol{\varepsilon}_{g,k}, \quad \boldsymbol{\varepsilon}_{g,k} \sim \mathcal{N}(\mathbf{0}, \mathbf{I}_K), \tag{41}$$

where $\lambda_k = -\frac{\alpha_{k+1} - \alpha_k}{\alpha_k\, h}$. After the update, renormalise $\mathbf{z}_g^{(k+1)}$ so that $\sum_i z_{gi}^{(k+1)} = 1$.

**Reverse Euler–Maruyama Step** Define the generator output:

$$\widehat{\mathbf{x}}_{g,\theta}^{(k)} = f_\theta(\mathbf{z}_g^{(k)}, t_k), s_\theta^{(k)} = \eta\left[\alpha_k\, \widehat{\mathbf{x}}_{g,\theta}^{(k)} - \mathbf{z}_g^{(k)}\right] - (\eta\alpha_0 - 1)\mathbf{1}. \tag{42}$$

Iterating backwards for $k = T, \ldots, 1$,

$$\mathbf{z}_g^{(k-1)} = \mathbf{z}_g^{(k)} + h\left[\lambda_k\left(\widehat{\mathbf{x}}_{g,\theta}^{(k)} - \mathbf{z}_g^{(k)}\right) - \frac{\lambda_k}{\eta}\, \Sigma\left(\mathbf{z}_g^{(k)}\right) s_\theta^{(k)}\right]$$

$$+ \sqrt{\frac{h\lambda_k}{\eta}}\, B_k\, \bar{\boldsymbol{\varepsilon}}_{g,k}, \quad \bar{\boldsymbol{\varepsilon}}_{g,k} \sim \mathcal{N}(\mathbf{0}, \mathbf{I}_K). \tag{43}$$

Initialize with $\mathbf{z}_g^{(T)} \sim \text{Dir}\left(\eta\alpha_T \mathbf{1}\right)$ and iterate to $\mathbf{z}_g^{(0)}$.

## C Description of Figures

During sampling we emulate Beta diffusion and generate two candidate outputs. The first, denoted by *out*, is a single–step prediction that depends only on the current timestep. The second, $out_1$,

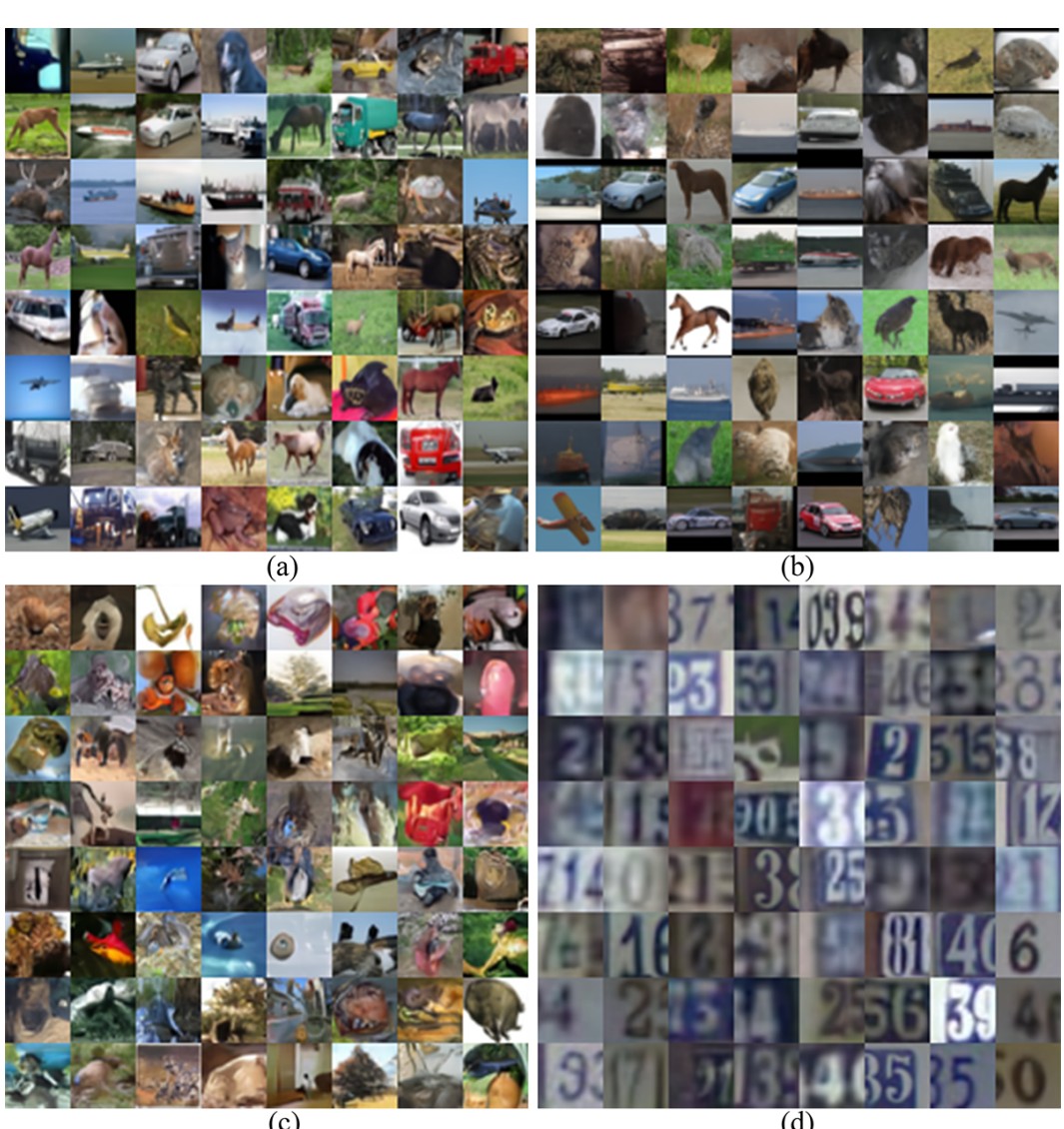

Figure 11: (a) CIFAR-10 images generated using GDD; (b) STL-10 images generated using GDD; (c) CIFAR-100 images generated using GDD; and (d) SVHN images generated using GDD. This figure demonstrates the effectiveness of GDD in producing high-quality images across diverse datasets.

---

**Algorithm 2:** Sampling of Grouped Dirichlet Diffusion

`// Input and Initialization Input:` Number of function evaluations (NFE) $J = 200$; generator $f_\theta$; timesteps $\{t_j\}_{j=0}^J$ with $t_0 = 0$ and $t_J = 1$ (or close to 1); scheduling $\alpha_{t_j}$ (beta linear/sigmoid).

**Initialization:**
- Set parameters: $S_{\text{shift}} = 0.6$, $S_{\text{scale}} = 0.39$, $c_{\text{start}} = 10$, $c_{\text{end}} = -13$, $\eta = 10000$, batch size $= 256$, $\omega = 0.97$, $\pi = 0.95$.
- Initialize $\hat{x}_0 = \mathbb{E}[\mathbf{x}_0] \times S_{\text{scale}} + S_{\text{shift}}$.

`// Scheduling Adjustment if` *NFE* $> 350$ **then**

$$\alpha_{t_j} = \frac{1}{1 + e^{-c_{\text{start}} - (c_{\text{end}} - c_{\text{start}})t_j}}.$$

**end**
**else**

$$\alpha_{t_j} = \left(\frac{1}{1 + e^{c_{\text{end}}}}\right)^{t_j}.$$

**end**

`// Sampling Procedure for` $j = J$ *downto 1* **do**

Sample
$$z_{t_j} \sim \text{GDD}\Big(\eta\, \alpha_{t_j}\, \hat{x}_0,\ \eta\big(1 - \alpha_{t_j}\, \hat{x}_0\big)\Big)$$

Compute
$$\hat{x}_0 = f_\theta\big(z_{t_j}, \alpha_{t_j}\big) \times S_{\text{scale}} + S_{\text{shift}}$$

Update
$$z_{t_{j-1}} = z_{t_j} + (1 - z_{t_j}) \times p(t_{j-1} \leftarrow t_j)$$

where
$$p(t_{j-1} \leftarrow t_j) \sim \text{GDD}\Big(\eta\,(\alpha_{t_{j-1}} - \alpha_{t_j})\,\hat{x}_0,\ \eta\Big(1 - (\alpha_{t_{j-1}} - \alpha_{t_j})\,\hat{x}_0\Big)\Big)$$

**end**

`// Output Return:` $\dfrac{\hat{x}_0 - S_{\text{shift}}}{S_{\text{scale}}}$.

---

refines this estimate by applying a nonlinear sigmoid transformation conditioned on the next–step parameter $\alpha_{\text{next}}$. Following qualitative inspection and quantitative assessment, we adopt *out* as the final generated image. Figure 9 juxtaposes the two predictions.

To demonstrate the capacity of GDD to synthesise high-fidelity images from complex datasets, we additionally provide uncompressed samples. Figure 7 presents $64 \times 64$ results on the CelebA dataset, whereas Figure 8 displays $128 \times 128$ outputs on the AFHQ dataset.

Figure 10 shows the training loss curve of GDD model. The horizontal axis denotes the cumulative number of images processed during training (in thousands), and the vertical axis shows the loss value. The blue curve represents the raw loss at each training step, the red curve depicts the moving average of the loss, and the shaded region corresponds to ±1 standard deviation. This figure illustrates that after an initial rapid decrease, the loss stabilizes and fluctuates within a narrow band, indicating convergence and improved training stability of the GDD model.

## D    COMPARISON WITH RECENT SIMPLEX-BASED GENERATIVE MODELS

**Hierarchical Expressiveness.** DDSM (Avdeyev et al., 2023a) and DFM (Stärk et al., 2024a) treat every sample as a single $K$-simplex, forcing all channels (or DNA bases) to compete globally and thus erasing local structure. Grouped Dirichlet Diffusion (GDD) instead divides the vector into independent Dirichlet groups—e.g., RGB channels, hyperspectral bands, or task-specific feature blocks. Each group follows its own concentration path, while a shared sigmoid schedule $\alpha(t)$ coordinates cross-group interaction, capturing fine-grained intra-group dependencies that sequence-level methods miss.

**Numerical Stability and Training Efficiency.** DDSM must integrate $K-1$ Jacobi SDEs whose diffusion terms explode near the simplex boundary; DFM learns a continuous flow that requires

an extra distillation pass for fast sampling. GDD injects multiplicative Dirichlet noise that stays tangent to the simplex, operates entirely in logit space, and minimizes a closed-form KL upper bound (KLUB). KLUB provides low-variance gradients, removes costly digamma terms, and avoids Jacobian or flow-matching penalties, yielding faster and more stable convergence.

**Scalability and Speed.** Because each group updates in parallel with a single Dirichlet draw, GDD scales linearly with the number of groups rather than vocabulary size. On CIFAR-10 it generates $\sim$43 images $s^{-1}$ in 200 sampling steps—matching DFM's distilled speed without a separate distillation stage—and achieves an FID of 2.76, outperforming both DDSM (image-agnostic) and DFM (DNA-focused).

GDD therefore generalizes Beta diffusion to high-dimensional, group-structured data while preserving closed-form dynamics and low-variance optimization. It combines the stochastic rigor of DDSM with the sampling speed of DFM and extends both beyond discrete sequences to multichannel images, spectrograms, and hierarchical histograms. These two algorithms (Algorithm 1 and Algorithm 2) respectively describe the training and sampling processes of Grouped Dirichlet Diffusion.

## E  THE USE OF LARGE LANGUAGE MODELS

We used a large language model (LLM) solely as a general-purpose writing assistance tool. Specifically, the LLM was employed to check spelling, correct grammatical errors, and improve the clarity and style of the manuscript text. The LLM did not contribute to the conception of the research ideas, the design of the methodology, the execution of experiments, or the analysis and interpretation of results. All scientific content, claims, and conclusions are solely those of the authors.

