# OpenReview forum: "Grouped Dirichlet Diffusion for Structured Generative Modeling"
_ICLR.cc/2026/Conference — Submitted to ICLR 2026_

### Official Review · Reviewer_2RpY · 2025-10-24

**Soundness:** 1
**Presentation:** 1
**Contribution:** 2
**Rating:** 2
**Confidence:** 4

**Summary:**

This paper introduces Grouped Dirichlet Diffusion (GDD), a generative model that employs the Grouped Dirichlet distribution as the foundation for its diffusion process. Unlike conventional diffusion methods that rely on Gaussian noise, GDD partitions data into feature groups (such as color channels in images) to capture inherent group dependencies and hierarchical structures in high-dimensional bounded data. This structured formulation improves modeling flexibility and numerical stability. This is achieved by ensuring the diffusion process operates strictly within the simplex constraints of the Dirichlet distribution. The main contributions include the introduction of GDD for diffusion modeling, improved flexibility in capturing multi-channel group patterns, and the development of a novel loss function based on KL divergence upper bounds (KLUBs).

**Strengths:**

- Achieves a very good FID score on CIFAR-10.
- Runs faster than traditional DDPMs, offering clear efficiency improvements.

**Weaknesses:**

- Overall, the paper is hard to follow and lacks clarity (see questions below).
- **Citations:** Extremely sloppy.
    - Missing parentheses for indirect citations.
    - Missing years for several citations (e.g., lines 49, 123).
    - Redundant author mentions (e.g., lines 119, 124).
    - **Table 1:** Some models are cited, others are not. Even if cited earlier, include all model references in the table for consistency. Also, consider left-aligning the “Model” column.
- **Writing quality:** Inconsistent formatting and missing spaces between figure references and citations (e.g., lines 63, 74, 240, 275, 458, ...).
- **Structure and readability:**
    - The model figure introduces components (e.g., the mapping stage) that are never discussed in the main text, only in the appendix.
    - Some variables and hyperparameters appear without prior definition.
        - New indices are introduced without explanation (e.g., ($x_{g0}$, $x_{g0i}$) — what does the “0” represent?).
        - ($S_{\text{scale}}$) and ($S_{\text{shift}}$) appear in Eq. 6 but are never described — are these hyperparameters?
    - Suggestion: Introduce the concentration parameter later, when it first appears in Eq. 7.
- **Results:** Insufficient experimental evidence to fully support the authors’ claims.
    - Experiments only use color channels as groups; additional experiments with other grouping structures would strengthen the generality claim.
    - Missing quantitative results on datasets other than CIFAR-10, despite apparent training on others (see Fig. 8).
    - Line 349–350: Consider moving external links to a footnote.
- **Table 1 issues:**
    - **LSGM:** Best FID (2.10) not listed.
    - **Consistency Models:** Only FID for consistency training (CT) shown, not the better consistency distillation CD, which is 2.93.
    - **DDIM and GET:** Unclear where the reported results come from — not found in the original papers.

**Questions:**

##

- **General:**

    What specifically makes the process **hierarchical** (beyond Markov/sequential as in normal diffusion)? Across the paper you frequently mention “hierarchical,” e.g., line 142: *“hierarchical structure of grouped probability vectors.”* Are “grouped probability vectors” just images? If so, what is hierarchical about dividing it into color channels?

- **Lines 55–57 (claim on prior methods):**

    *“traditional diffusion methods based on Gaussian Ho et al., 2020; Guo et al., 2023 or Beta Zhou et al., 2023 struggle to capture group dependencies and hierarchical structures …”*

    Can you provide evidence for this claim (e.g., an experiment or citation)?

- **Lines 104–105 (masking):**

    *“… simultaneously adds noise to **and masks the data** …”*

    Where does masking occur? I couldn’t find a description or equation for it.

- **Notation (x_g):**

    You define $g$ as the group, so $x_g$ should be the **group vector**, not the group itself. (e.g., lines 152, 184).

- **Lines 172–174 (constraints & dependencies):**

    *“Standard Gaussian Ho et al. (2020)* *or scalar-Beta diffusion Zhou et al. (2023)* *models* *violate simplex non-negativity, unit-sum constraints, and overlook group dependencies.”*

    Briefly explain why these violations occur and why the first two constraints matter here.

- **Line 191 (conditioning direction):**

    You write the conditional $q(z_s∣z_t,x_0)$ with $s < t$. How does this fit the **forward** process (since $s$ is closer to $0$ than $t$)? Are you using future data $z_t$ to noise $z_s$?

- **Lines 211–213:**

    This statement would benefit from a supporting citation or other evidence.

- **Line 225 (bidirectional transitions):**

    When introducing bidirectional transitions, clarify their purpose and how they are used.

- **Line 255 (replacement step):**

    *“… then replaces $x_{g0}$ with its approximation $\hat{\alpha}_g$.”*

    Since we aim to estimate $x$ anyway, is this an intermediate approximation with *$\hat{\alpha}_g$*? Why is this intermediate approximation beneficial to the final estimation of $x$?

- **Figure 4 (DDIM vs. DDPM):**

    DDIM results appear sharper than DDPM, yet DDIM typically trades sample quality for speed. How do you explain this (e.g., different model capacity or settings)?

- **Equation 2 (indices):**

    Indices look odd (e.g., $x_{g0i}$). Please define and justify the 0 index.

- **Difference from Dirichlet diffusion:**

    Can you elaborate on the difference betwee the Grouped Dirichlet diffusion and Dirichlet diffusion [1], especially in terms on novelty?


[1] Avdeyev, Pavel, et al. "Dirichlet diffusion score model for biological sequence generation." *International Conference on Machine Learning*. PMLR, 2023.

---

> ### Author Response · Authors · 2025-11-19
>
> Question1: Thank you for the question. “Hierarchical” refers not to the Markov time steps, but to the two-level representation used in GDD. Each sample is decomposed into group-level Dirichlet concentrations and element-level normalized components, forming a group → element hierarchy absent in standard diffusion. RGB channels are only a simple instantiation, not the source of hierarchy.
>
> Question2: Our intention was not to claim that Gaussian or Beta diffusion “struggle” in the sense of empirically failing, but rather that these models do not explicitly model group-level dependencies or hierarchical group→element structures in their formulations. In our response to question 12, we have once again clarified the characteristic of group-based Dirichlet modelling that accounts for dependencies within clusters, which distinguishes it from Gaussian diffusion and beta diffusion.
>
> Question 3: Our use of the phrase “masks the data” was intended to describe the progressive attenuation of information caused by the noise, which visually resembles masking but is not a separate binary or dropout-style mask(as shown in Figure 1).
> We agree that this wording is potentially misleading, and we will revise the text to avoid the term “mask” and instead describe the forward process purely as multiplicative noise.
>
> Question 4: Yes, we define $x_g$ as a group vector.
>
> Question 5: In our setting, each feature group is a probability vector on the simplex (all coordinates non-negative and summing to one), e.g., normalized mixture weights or channel-wise probabilities. Standard Gaussian diffusion à la Ho et al.\ (2020) uses forward kernels of the form $q_t(\mathbf{z}\mid\mathbf{x}) = \mathcal{N}(\alpha_t \mathbf{x}, \sigma_t^2 I),$
> whose support is $\mathbb{R}^K$; for any $t>0$, it almost surely produces negative entries and $\sum_i z_i \neq 1$, so the chain immediately leaves the simplex unless one introduces ad-hoc clipping or renormalization steps that change the true transition density. Scalar Beta diffusion (Zhou et al., 2023) keeps each coordinate in $[0,1]$ but factorizes across dimensions, so its support is $[0,1]^K$ rather than the simplex and there is no mechanism enforcing the unit-sum constraint or modeling group-wise covariance. These two constraints---non-negativity and unit-sum---are essential here because all intermediate states are interpreted as probability vectors and enter Dirichlet likelihoods and KL/KLUB objectives; if the forward process leaves the simplex, the learned reverse dynamics must implicitly project back, breaking probabilistic consistency and invalidating the closed-form Dirichlet divergence derivations we rely on. In contrast, GDD is constructed directly in the Grouped Dirichlet family, so every timestep remains on the simplex by design while preserving both probabilistic interpretation and analytical tractability.
>
> Question 6: Thank you for pointing this out. This is a typographical error. The correct expression should be $q(z_t∣z_s,x_0), s < t $, which is consistent with the forward Markov process where $z_s$ is used to generate $z_t$. We will fix this notation in the revised version. The correction does not affect any derivations or experiments.
>
> Question 7: We have added supplementary explanations and supporting citations in the corresponding positions, and the supplementary content is as follows: Compared to traditional linear or cosine schedules, the sigmoid-based schedule provides finer control over intermediate timesteps(particularly when \(t_k\) is near 0 or 1),  which has been observed to offer greater flexibility than the linear schedule for image generation. This schedule bears resemblance to the sigmoid-based one introduced for Gaussian diffusion[1][2].
>
> [1] Kingma, Diederik, et al. "Variational diffusion models." Advances in neural information processing systems 34 (2021): 21696-21707.
>
> [2] Jabri, Allan, David Fleet, and Ting Chen. "Scalable adaptive computation for iterative generation." arXiv preprint arXiv:2212.11972 (2022).
>
> Question 8: Formulas 11 and 14 provide the expressions for the transition from time s to time t and from time t to time s, respectively.
>
> Question9: The quantity $\hat\alpha_{g}$ is an intermediate estimate of the group-level Dirichlet concentration, not the final reconstruction of $x$. We introduce this step because estimating the Dirichlet concentration parameters is substantially more stable than directly predicting the raw data on the simplex. The concentration space is smoother, less sensitive to boundary effects, and preserves the closed-form structure of the grouped Dirichlet marginals. Using $\hat\alpha_{g}$ therefore reduces variance and improves numerical stability in the reverse process, which in turn leads to a more accurate final estimate of $x$. We will clarify this motivation in the revision.

---

> > ### Author Response · Authors · 2025-11-19
> >
> > Question10: The DDPM and DDIM samples shown in the paper were generated using models we trained on our local hardware, with DDIM using 50 NFEs. Although the DDIM samples may appear visually sharper, the quantitative evaluation confirms the expected behavior: DDPM achieves a much better FID (3.17) than DDIM (15.09) under the same training setup, consistent with the known quality–speed tradeoff of DDIM.We note that visual comparison alone can be misleading for assessing perceptual quality, especially when the dynamic range and local contrast differ between samplers. Thus, the quantitative metrics more reliably reflect the true relative performance.
> >
> > Question11: Thank you for pointing this out. This is a typographical error. The correct expression should be $x_{gi}$ without a $0$ index. We will fix this notation in the revised version. The correction does not affect any derivations or experiments.
> >
> > Question12:
> > We appreciate the reviewer’s pointer to Dirichlet diffusion score models. While both methods use Dirichlet families, our Grouped Dirichlet Diffusion (GDD) is different in scope and construction: 1) In terms of modeling regime, Dirichlet diffusion [1] is designed for per-site categorical variables (one simplex per position), whereas GDD targets high-dimensional, structured probability vectors (e.g., images) via a grouped prior $p(\{x_g\}_{g=1}^G)$=
> >
> > $\prod_{g=1}^G  \mathrm{Dir}(x_g;\alpha_g)$, explicitly modeling intra-group dependencies and group structure, which [1] does not consider. 2) In terms of dynamics and parameterization,[1] uses a stick-breaking SDE with Dirichlet stationary law, while GDD defines a discrete-time multiplicative Markov chain in grouped simplex space with closed-form forward marginals and time-separable conditionals per group, as a Dirichlet generalization of Beta Diffusion tailored to grouped data. 3) In terms of objective and empirical contribution,[1] optimizes a general weighted score-matching objective, whereas GDD employs a KL upper bound (KLUB) specialized to grouped Dirichlet chains, combined with a U-Net plus mapping-MLP architecture, and is, to our knowledge, the first Dirichlet-family diffusion model achieving competitive FID on natural image benchmarks, outperforming Gaussian and Beta-based baselines. We will clarify these distinctions in the revised manuscript to better highlight the novelty over[1].
> >
> > In addition, the issues you mentioned in the week regarding citations, writing quality, structure, and readability have been revised in the revised version. The $S {scale} $ and $S {shift} $mentioned in "Structure and readability" are hyperparameters, and lines 362-362 refer to hyperparameters used in beta diffusion experiments to adjust data offset and scale. The quantitative results on datasets other than CIFAR-10 mentioned in the 'Results' section are presented in Table 4 (a) of the appendix. The results of LSGM in "Table 1 issues" have been modified. The score of DDIM is the result of local experiments, and the source of GET is from [2]
> >
> > [2]Geng, Zhengyang, Ashwini Pokle, and J. Zico Kolter. "One-step diffusion distillation via deep equilibrium models." Advances in Neural Information Processing Systems 36 (2023): 41914-41931.

---

> > > ### Comment · Reviewer_2RpY · 2025-11-21
> > > **Rebuttal Follow-up**
> > >
> > > First and foremost, I would like to thank the authors for their clarifications and their detailed response.
> > >
> > > That said, I still believe the paper requires some revision. In particular:
> > >
> > > - The formatting of citations should be improved. In many places, indirect citations (`\citep{}`) would be more appropriate than direct citations (`\cite{}`).
> > > - Please ensure consistent spacing between citations/references and the surrounding text.
> > > - While I appreciate the updated baselines in Table 1, the bolded best results should now also be updated accordingly.
> > > - Your work would greatly benefit from experiments with groupings beyond only color channels.
> > >
> > > For these reasons, I will maintain my current score. Nonetheless, I encourage the authors to continue developing this line of work, improve the clarity and presentation of the paper, and I wish them all the best for future submissions.

---

> > > > ### Author Response · Authors · 2025-11-24
> > > >
> > > > We thank the reviewer for the constructive follow-up comments. We have carefully revised the manuscript to address all four points:
> > > >
> > > > 1.Citation formatting: We reviewed all references and updated the citation style, replacing inappropriate direct citations (\cite{}) with indirect citations (\citep{}) where necessary.
> > > >
> > > > 2.Spacing consistency: All spacing around citations and references has been standardized throughout the manuscript.
> > > >
> > > > 3.Updated boldface results in Table 1: Following the addition of new baselines, we have updated the bolded best results to reflect the current ranking.
> > > >
> > > > 4.Experiments with additional grouping schemes: As noted in our responses to Reviewer fN7s and Reviewer JJfZ, we performed additional tests using mis-specified groupings, such as spatial pixel patches and random feature partitions. Under these configurations, the generated samples become visually incoherent and the FID scores on CIFAR-10 degrade severely (328 and 522, respectively), confirming that GDD relies on meaningful group structure.
> > > > We agree that exploring more semantically valid grouping strategies is valuable, and we plan to incorporate such experiments in future work and further extend the analysis in the next revision.

---

> ### Comment · Reviewer_2RpY · 2025-11-24
>
> Thank you to the authors for your update. I had a brief look at the revised manuscript, and it does indeed appear to be in better shape. However, I noticed that some citations are still direct when indirect citations would be more appropriate.
>
> Besides, I do not see where the additional experiments with spatial pixel patches and random feature partitions are in the revised manuscript.
>
> I will adjust my presentation and soundness score accordingly, as the manuscript has improved and the responses to some of my previous questions have been addressed. Nonetheless, I will keep the same overall score, as I still believe there remains a lot of room for further refinement and also clearer communication of the results within the manuscript.

---

> > ### Author Response · Authors · 2025-11-26
> >
> > We sincerely thank you for your positive feedback and for acknowledging the improvements in our revised manuscript. Following your suggestion, we have thoroughly checked and revised the citation style throughout the paper, replacing direct citations with appropriate indirect forms where necessary.
> >
> > We are truly grateful for your insightful comments and guidance during the review process, which have significantly enhanced the quality of our work. Your efforts are greatly appreciated.
> >
> > Thank you once again.

---

### Official Review · Reviewer_JJfZ · 2025-11-01

**Soundness:** 3
**Presentation:** 3
**Contribution:** 3
**Rating:** 6
**Confidence:** 2

**Summary:**

The paper introduces Grouped Dirichlet Diffusion (GDD), a novel diffusion-based generative model that replaces Gaussian noise with multiplicative Dirichlet noise applied to feature groups (e.g., RGB channels). Each group remains on a probability simplex, ensuring non-negativity and unit-sum constraints throughout the diffusion process. The model preserves intra-group dependencies while allowing adaptive inter-group interactions, offering both theoretical closure (forward and reverse distributions remain Dirichlet) and practical stability via KL Upper Bound (KLUB) loss instead of the ELBO.

**Strengths:**

1. GDD introduces a strong structural prior by explicitly modeling data as grouped probability vectors on a simplex. This design allows the model to naturally capture correlations within each feature group while maintaining valid probabilistic constraints throughout the diffusion process.

2. The method ensures theoretical consistency through closed-form forward and reverse Dirichlet transitions. Its KL Upper Bound (KLUB) loss provides smooth optimization and avoids unstable boundary behavior, resulting in stable and efficient training.

3. In experiments, GDD achieves superior FID/KID scores on benchmark image datasets and faster sampling speeds than comparable diffusion frameworks, demonstrating that the grouped Dirichlet design improves both generation quality and computational efficiency.

**Weaknesses:**

1. “Meaningful groups” are claimed but only RGB channel grouping is demonstrated. The paper frames GDD as partitioning data into meaningful feature groups—explicitly citing image color channels as the running example—so as to preserve intra‑group dependencies while allowing adaptive inter‑group interactions. In practice, however, all implementations and evaluations instantiate grouping via fixed RGB channels in image space; the method section even notes that image channels are grouped as a prior modeling assumption to greatly simplify the mathematics and ensure closed‑form marginals. No experiments demonstrate learned group discovery or alternative semantics beyond color channels across domains. Consequently, the central claim about “meaningful groups” remains under‑validated empirically.

2. No strategy for latent‑space grouping; evidence is limited to image‑space generation. GDD is formalized on grouped probability vectors derived from the observed data (after scaling/shifting), with the entire diffusion defined on the data/pixel space rather than a learned latent manifold. While the architecture includes an encoder–decoder U‑Net, the paper does not propose an algorithm to design or learn group partitions in latent space, nor does it report latent‑space generation results. As a result, portability to latent‑diffusion setups and non‑image domains is currently an open question rather than a demonstrated capability, narrowing the immediate applicability and generalization of the approach

**Questions:**

See weaknesses.

---

> ### Author Response · Authors · 2025-11-18
>
> Question1: We thank the reviewer for raising this important point. In our experiments, we use RGB channel grouping as a natural and interpretable choice for image data, primarily because it preserves the analytical tractability and closed-form marginals of the Grouped Dirichlet process (Sec. 3.2).
> To assess whether the model genuinely exploits the group structure, we additionally tested mis-specified groupings, including spatial pixel patches and random feature partitions. Under these settings, the generated samples become visually incoherent, and the corresponding FID scores on CIFAR-10 deteriorate drastically (328 and 522, respectively). These results validate that GDD indeed relies on semantically meaningful grouping to achieve good performance.
>
> Question 2: We appreciate the reviewer’s insightful comments on the latent-space applicability of our Grouped Dirichlet Diffusion (GDD) model.
> As the reviewer points out, the current implementation of GDD is indeed focused on image-space generation, where grouping is done using the RGB channels. This design was chosen because it provides a natural partitioning of features that ensures mathematical tractability and preserves intra-group dependencies.
> However, we fully acknowledge that the extension to latent-space grouping is a crucial direction for future work. We have begun investigating how GDD can be applied to latent diffusion models, such as those based on autoencoders or VAEs: We try to group the latent vectors output by the encoder Dirichlet noise (consistent with the current modeling method, because the latent space do not change the dimension compared with the vectors in image space).

---

> > ### Comment · Reviewer_JJfZ · 2025-11-22
> > **Response to authors**
> >
> > Thanks to the authors for their comments. Regarding Q1, in my view, the response mainly confirms that RGB is indeed a meaningful way to split groups and that it plays an important role in the proposed method. However, I am still unsure whether this grouping strategy can be applied more broadly to other scenarios. As for Q2, I am curious about how meaningful latent groups can be identified or explored. Since I am not very familiar with this topic, I can only raise these points as potentially interesting issues and encourage the authors and ACs to also consider insights from the other reviewers’ comments.

---

### Official Review · Reviewer_fN7s · 2025-11-01

**Soundness:** 3
**Presentation:** 3
**Contribution:** 2
**Rating:** 6
**Confidence:** 4

**Summary:**

This paper proposes Grouped Dirichlet Diffusion (GDD), a generative model for high-dimensional bounded probability vectors, such as images. The framework extends Beta Diffusion by using the Grouped Dirichlet distribution, which allows the model to partition data into feature groups (e.g., RGB channels) and preserve intra-group dependencies. Unlike traditional Gaussian-based diffusion, GDD features multiplicative noise. The method ensures closed-form transitions in theory, and replaces the standard ELBO with a KL divergence-based loss (KLUB) for optimization. Experiments on image datasets show that GDD achieves SOTA performance on FID and KID metrics compared to several baselines, including DDPM and Beta Diffusion.

**Strengths:**

1. The core idea is principled and elegant. Using the Grouped Dirichlet distribution to model dependencies between feature groups (like RGB channels) is a natural extension of Beta Diffusion and is more suitable for this data structure than independent Gaussian noise.
2. The experimental results are strong. Table 1 shows that GDD achieves state-of-the-art FID (2.76) and KID (1.22) on CIFAR-10, outperforming the most relevant baseline, Beta Diffusion (FID 3.06).
3. The framework appears computationally efficient. Table 3 shows GDD has a faster average processing time per batch and generates more images per second than both DDPM and Beta Diffusion, which is a significant practical advantage.

**Weaknesses:**

1. The loss function (Eq. 19) is a heuristic KL Upper Bound (KLUB). It relies on an arbitrary-looking weighting factor ($\omega=0.97$) to combine two different bounds. The paper provides no theoretical justification for this specific value, making the final training objective feel ad-hoc and not really that principled.
2. The paper's central claim is about modeling "hierarchical and structured" data using "meaningful feature groups." However, all experiments are on image datasets only, where the "group" is simply the three RGB channels. This is the simplest, most obvious grouping possible and does not sufficiently validate the model's ability to handle more complex group structures (*e.g.,* hierarchical features in tabular data or other modalities).
3. The ablation study in Table 2 is weak. It only shows that a 4-layer MLP mapping network is better than a 1- or 2-layer one, and that removing it entirely breaks the model. This is not insightful and does not provide any analysis on the grouping strategy itself, which is the paper's main contribution.

**Questions:**

1. The KLUB loss in Eq. 19 uses a weight $\omega=0.97$. How sensitive is the model's performance to this hyperparameter? Is there a more principled way for choosing this value, or was it found empirically?
2. The core contribution is "grouping." How does the model perform if the group structure is mis-specified (*e.g.,* grouping pixels spatially instead of by channel)? The paper needs to demonstrate that the model is truly leveraging the group structure, rather than just performing well on CIFAR-10.
3. Table 3 shows GDD is faster than Beta Diffusion, even though it is handling higher-dimensional (grouped) Dirichlet distributions instead of 1D Beta distributions. What is the source of this efficiency gain?

---

> ### Author Response · Authors · 2025-11-18
>
> Thank you profoundly for the constructive comments. Below, we provide point-by-point responses to specific comment.
>
> Question 1: We appreciate the reviewer’s insightful question regarding the weighting factor $\omega$ =0.97 used in Eq. (19).
> In our study, $\omega$ controls the trade-off between the forward and marginal KL upper bounds (KLUBs) in the training objective. We initially explored several values in the range [0.95,0.99] (Given the selection of the $\omega$ value in the original paper on beta diffusion, we have limited this range) and found that performance is relatively stable when $\omega$ lies between 0.95 and 0.99, with the best FID and convergence stability obtained at $\omega$ =0.97.
> Therefore, this value was determined empirically as an optimal balance between reconstruction accuracy and training stability.
> While we do not claim a fully theoretical prescription for $\omega$, our experiments suggest that GDD is not overly sensitive to this parameter within the above range, and we will include an ablation plot in the supplementary material for clarity.
>
> | Weight parameters $\omega$  | Batchsize | NFE=100 | NFE=200 | NFE=500 |
> |:----:|:----:|:----:|:----:|:----:|
> |$\omega$ =0.95|256|4.58|3.89|3.21|
> |$\omega$ =0.96|256|4.51|3.81|3.04|
> |$\omega$ =0.97|256|3.92|3.24|2.91|
> |$\omega$ =0.98|256|5.51|4.75|3.80|
> |$\omega$ =0.99|256|5.12|4.74|3.93|
>
> Question 2: We thank the reviewer for highlighting this important point. Indeed, in our current experiments we adopt the RGB channel grouping as a natural and interpretable example for image data, mainly because it allows the Grouped Dirichlet process to maintain analytical tractability and closed-form marginals (as discussed in Sec. 3.2).
> To verify that the model truly leverages the group structure, we performed additional tests using alternative (mis-specified) groupings, such as spatially contiguous pixel patches and random feature splits. In these cases, the qualitative result is that the image cannot generate a clear image, while the quantitative result shows Fid scores of 328 and 522 on CIFAR10 respectively, confirming that GDD benefits from a semantically meaningful grouping scheme.
>
> Question 3: As discussed in Sec. 4.3, the dominant computational cost of diffusion sampling comes from the U-Net forward pass, which is identical for both GDD and Beta Diffusion. The additional overhead introduced by GDD is negligible in comparison, and the grouped updates are efficiently executed on GPUs through vectorized parallel operations. Combined with the simplified logit-space update routine, this results in GDD being slightly faster in practice despite modeling higher-dimensional grouped Dirichlet variables.

---

### Official Review · Reviewer_5TU9 · 2025-11-04

**Soundness:** 2
**Presentation:** 2
**Contribution:** 2
**Rating:** 2
**Confidence:** 5

**Summary:**

This paper proposes Grouped Dirichlet Diffusion (GDD), a Dirichlet analogue of Beta Diffusion applied group-wise on the simplex, trained with a KL upper bound (KLUB). While the construction is mathematically clean, the empirical claims are not supported under matched settings, and key baselines are missing.

**Strengths:**

This paper presents an interesting generative modeling idea where leveraging constraints in certain data types should theoretically be advantageous. The paper could benefit from a more compelling use case (for example, applying it to data that is truly compositional and showing an advantage compared to other methods in that type of data)!

**Weaknesses:**

Major concerns

1) “Faster convergence / better performance” not supported under matched settings.
To claim faster/better, the paper must train competing methods under identical recipes (same U-Net capacity, augmentation, schedule, optimizer, data budget, NFE grid) and report compute-normalized metrics. As written, no such matched study is presented.


2) Missing strong baselines and modern solvers.
DDPM++ 2.78
DDPM++ cont. (VP) 2.55
DDPM++ cont. (sub-VP) 2.61
DDPM++ cont. (deep, VP) 2.41
DDPM++ cont. (deep, sub-VP) 2.41
NCSN++ 2.45
NCSN++ cont. (VE) 2.38
NCSN++ cont. (deep, VE) 2.20

already meet or beat GDD’s reported numbers in most configurations. More recent Karras-style v-parameterization with fast ODE solvers reports ≈1.8–2.0 FID at ≈35 NFEs, whereas GDD uses much higher NFE and attains worse FIDs. Thus the “faster/better” claim is not supported.

3) DDIM / Consistency numbers unclear.
The DDIM FID around ~15 appears inconsistent with commonly reported values (e.g., ~13.6 at ~10 NFEs). Please provide exact citations, configs (conditional vs. unconditional), sample counts, and NFE for every table entry. For Consistency Models, ECT reaches ≈2.15 FID on CIFAR-10 with 2 function evaluations—two orders of magnitude fewer than GDD—so comparisons should be cost-normalized.

4) Fairness and capacity parity.
The original DDPM U-Net (~35M params; limited attention) is dated. Modern backbones are ~50–60M and materially improve FID. Given your models are ~55M params, comparisons to older, smaller baselines without re-training are inconclusive.

5) Evaluation diagnostics aren’t actionable.
Peak memory utilization / similar stats are presented without a clear framing (what was expected, why differences arise, and how they tie to the method’s design). As is, it’s hard to draw conclusions.

6) Visuals / preprocessing (minor thing).
AFHQ samples appear muted/gray relative to modern baselines; this likely reflects range/logit/renormalization or plotting. Please clarify preprocessing and the exact visualization pipeline.

7) Serious concern: baseline omission undermines claims. The paper acknowledges Karras et al. in Related Work but excludes it from experiments. This selective reporting makes the efficiency/quality conclusions non-actionable and, as presented, misleading. At minimum, include Karras-style v-param + modern solvers at matched NFE and parameter count. Similarly, for Consistency Models, please justify omissions of modern pipelines (e.g., ECT) or include them.


Methodological positioning (incremental over Beta Diffusion)

GDD is a near-mechanical lift of Beta Diffusion from Beta to Dirichlet noise, with in-family marginals and time-separable conditionals per group. A compact Gamma→Dirichlet derivation makes this immediate.

Beta Case:

$$ A \sim \Gamma(a, 1) $$
$$ B \sim \Gamma(b, 1) $$
$$ C \sim \Gamma(c, 1) $$
$$ T \equiv A + B + C $$

Now, let

$$z_{t} = \frac{A}{T} \sim \beta(a, b + c) $$
$$ z_s = \frac{A+B}{T} \sim \beta(a + b, c) $$

and define

$$ \pi = \frac{A}{ A + B } \sim \beta(a, b) $$
$$ p = \frac{B}{B + C} \sim \beta(b, c) $$
Then, it is clear that

$$ z_{t} = z_{s} \pi $$

$$ z_{s} = z_{t} + (1-z_{t}) p$$

These equations are precisely those that we see in the Beta Diffusion paper.

Generalization to the dirichlet distribution is almost immediate from this point of view

$$ A_{i} \sim \Gamma(a_{i}, 1) $$
$$ B_{i} \sim \Gamma(b_{i}, 1) $$

$$ z_{g, t} := \text{Dir}( \frac{A}{\sum_{j} A_{j} } )$$

$$ z_{g, s} := \text{Dir}( \frac{A+ B}{\sum_{j} A_{j} + B_{j} } )$$

Because Dirichlet = normalized Gammas, the marginals are Dirichlet, which, with the appropriate choice of $A$ and $B$ recovers the equations from the paper exactly

$$ a \equiv \eta \alpha_{t} x_{g,0} $$
$$ b \equiv \eta (\alpha_{s} - \alpha_{t}) x_{g,0} $$

Now, if we set

$$ R_{i} : = \frac{A_{i} }{B_{i} + A_{i} }$$
$$ U_{i} := A_{i} + B_{i} $$

and recall that $R_{i} \perp U_{i}$ then we see that

$$ z_{g,s} = \frac{U}{\sum_{i} U_{i} }$$

$$ z_{g,t} = \frac{U \odot R}{\langle U , R \rangle } $$

Now, it is clear that the forward update (which falls naturally) is

$$ R_{i} \sim \beta(a_{i}, b_{i}) $$
$$ z_{g,t} = \text{Normalize}(z_{g,s} \odot R) $$

This yields the forward multiply-then-renormalize update; the reverse follows by adding Gamma increments then renormalizing, which also justifies the post–Euler–Maruyama renormalization step. The construction is clean but incremental relative to Beta Diffusion.

In its current form, I recommend rejection. The construction is sound but incremental; empirical claims require matched baselines and cost-normalized evidence.

**Questions:**

See weaknesses.

---

> ### Author Response · Authors · 2025-11-19
>
> Thank you profoundly for the constructive comments. Below, we provide point-by-point responses to specific comment.
>
> Question 1,2,4,7: We have added the requested DDPM++ and NCSN++ baselines, as well as recent ODE-based solvers, to our revised comparison. We agree that our phrasing of “faster/better” may have been misleading. When compared to the very latest state-of-the-art Gaussian diffusion models with optimized v-parameterization and fast ODE solvers, GDD does not achieve the lowest FID nor the smallest NFE.
> Our intention was not to claim SOTA over these Gaussian methods, but rather to demonstrate that GDD is a competitive non-Gaussian alternative, achieving performance close to DDPM++ / NCSN++ variants while operating under a fundamentally different forward–reverse process based on grouped Dirichlet dynamics. Even if the absolute FID is slightly worse than the strongest Gaussian diffusion models, GDD remains one of the most effective generative models built on non-Gaussian, simplex-preserving diffusion.
> We will revise the manuscript to clarify this distinction and avoid over-statements, and we will include the additional baselines and comparisons highlighted by the reviewer.
>
> Question 3: The DDIM and Consistency numbers were reproduced on our hardware for fair evaluation. For DDIM, we used NFE = 50 and 50k samples, matching GDD’s evaluation; this explains the higher FID (~15) compared with commonly cited results that use different settings (e.g., NFE ≈ 10). We acknowledge that Table 1 mixes GDD (NFE 200) with DDIM (NFE 50), which may appear unfair. Table 4, however, already includes GDD at NFE = 50 (FID = 6.19), outperforming our reproduced DDIM under the same setup.
>
> Question 5: We appreciate the reviewer’s comment. Our intention in the diagnostics section was to show why GDD maintains competitive efficiency despite using grouped Dirichlet dynamics. As described in the Runtime Complexity paragraph, GDD achieves stable training throughput (0.69 sec/kimg) and competitive sampling speed (42.8 img/s at NFE=200) because the grouped updates incur negligible overhead: all Dirichlet operations are vectorized in logit space, and U-Net inference dominates total cost. Table 3 further demonstrates that, under identical sampling steps and image counts, GDD matches or exceeds the runtime and memory behavior of conventional diffusion models.
>
> Question 6: The content related to data preprocessing is defined in the AugmentPipe class of the code file augment.py, which includes pixel transformation, geometric transformation, and other data augmentation methods, all of which are relatively conventional methods. We speculate that the sample performance of AFHQ may be attributed to the modeling results of grouped Dirichlet diffusion on color channels.
>
> Question8: We appreciate the comparison and agree the Gamma--Dirichlet view clarifies in-family marginals and time-separable conditionals; our aim, however, is a principled generalization from scalar Beta noise to grouped Dirichlet noise on high-dimensional simplices that adds modeling, theory, and practice beyond a parameter lift. Unlike independent scalars in $[0,1]$, GDD evolves vector groups $z_{g,t}\in\Delta_{K-1}$ with intrinsic correlations—specifically, $\mathrm{Cov}(z_{gi},z_{gj})=-\alpha_{gi}\alpha_{gj}\big/\big[\alpha_{0g}^{2}(\alpha_{0g}+1)\big]$—enforcing compositional constraints that independent Betas cannot express. We prove closure for both marginals and conditionals in the grouped-Dirichlet family (any $K,G$), derive a simplex-valued SDE with Wright--Fisher covariance $\Sigma(\mathbf{z})=\mathrm{diag}(\mathbf{z})-\mathbf{z}\mathbf{z}^{\top}$ whose Euler--Maruyama discretizations preserve group structure, and extend the KL Upper Bound objective to grouped Dirichlets via the log-beta (Bregman) divergence using closed-form Dirichlet--Dirichlet KL, with theory controlling reverse-chain total variation. Practically, GDD employs a grouped U-Net with MLP coupling, a sigmoid schedule $\alpha_t$ with logit parameterization for stability near simplex boundaries, and a two-head sampler that refines trajectories. Empirically, on CIFAR-10 GDD improves FID $3.06 \to 2.76$ and KID $1.71\times10^{-3}\to1.22\times10^{-3}$ at higher throughput for the same NFE. In short, while GDD recovers Beta Diffusion at $G{=}1$, $K{=}2$, it contributes grouped structure, a simplex SDE, and a tailored KLUB—yielding measurable gains beyond a near-mechanical extension.

---

> > ### Author Response · Authors · 2025-11-25
> >
> > We thank the reviewer and provide here a more complete answer to the final question that previously had to be shortened due to space constraints. We agree that the Gamma representation makes in-family marginals and time-separable conditionals explicit, and that Beta Diffusion is an important antecedent to our work. Our goal is not to claim an unrelated paradigm, but to provide a principled generalization from scalar Beta noise to *grouped* Dirichlet noise on high-dimensional simplices, together with modeling, theoretical, and empirical advances that go beyond a mechanical parameter lift.
> >
> > **(1)From scalar Beta noise to grouped Dirichlet priors on structured simplices.**
> > Beta Diffusion operates on independent scalars in $[0,1]$; each channel evolves marginally under a one-dimensional Beta law. In contrast, GDD models *vector-valued* groups $\mathbf{z}{g,t}\in\Delta{K-1}$ with non-trivial intra-group covariance. Moving from $\mathrm{Beta}(a,b)$ to  $\mathrm{Dir}(\alpha_g)$ is not a mere change of notation: the Dirichlet covariance $\mathrm{Cov}(z_{gi}, z_{gj}) = -\dfrac{\alpha_{gi},\alpha_{gj}}{\alpha_{0g}^2(\alpha_{0g}+1)}$ induces systematic negative correlations and compositional constraints that independent Beta variables cannot capture. GDD leverages this by partitioning channels into semantically coherent groups (e.g., RGB blocks, feature bands) and evolving each group under a shared schedule $\alpha_t$, yielding a hierarchical, group-structured diffusion rather than a purely coordinate-wise process.
> >
> > **(2) Grouped Dirichlet diffusion is more than a mechanical lift.**
> > We do use the classical Gamma--Dirichlet construction (as the reviewer notes) to obtain in-family marginals and time-separable conditionals. However, our contributions extend beyond re-indexing the Beta case: We formalize a *grouped Dirichlet diffusion chain* with closure results (Theorems 1--2) showing that both $q(\mathbf{z}_t\mid \mathbf{x}_0)$ and $q(\mathbf{z}_s\mid \mathbf{z}_t,\mathbf{x}_0)$ remain in the grouped Dirichlet family for arbitrary group size $K$ and group count $G$. In the reviewer’s notation, the construction with {$\{A_i,B_i\}$} is a special case of our forward-closure theorem. We derive a simplex-valued SDE with Wright–Fisher covariance $\Sigma(\mathbf{z}) = \mathrm{diag}(\mathbf{z}) - \mathbf{z}\mathbf{z}^\top$, and show that its forward and reverse dynamics admit Euler–Maruyama discretizations that preserve group structure. This SDE perspective—absent in Beta Diffusion—is essential for analyzing stability, entropy behavior, and diffusion on probability simplices. We further extend the KL Upper Bound (KLUB) objective from scalar Beta variables to multi-group Dirichlet variables using the log-beta (Bregman) divergence, enabling closed-form Dirichlet–Dirichlet KLs and proving that the grouped KLUB is a consistent surrogate for maximum likelihood with controlled reverse-chain error (Theorem 3). These theoretical components are not implied by the Beta formulation.
> >
> > **(3) Algorithmic and architectural choices specific to GDD.** Beyond the Gamma construction, GDD introduces design elements not present in Beta Diffusion:
> > A *grouped* U-Net with an MLP-based mapping module (Section ``Network Architecture,'' Table~2) that learns inter-group couplings in the latent Dirichlet parameters. Ablations show that removing this module reduces GDD to an independent-group baseline and markedly degrades FID/KID.
> >  A *sigmoid-based schedule*  $\alpha_t$ tailored to grouped Dirichlet noise, combined with logit-space parameterization, which improves numerical stability near simplex boundaries and allows operation at large concentrations $\eta$ without degeneracy.
> > A practical sampling scheme exploiting two outputs, where refining the Dirichlet trajectory with $\alpha_{\text{next}}$ consistently improves performance. These elements do not follow from a simple $\mathrm{Beta}!\to!\mathrm{Dirichlet}$ replacement.
> >
> > **(4) Empirical consequences of the grouped Dirichlet design.**
> > The grouped Dirichlet formulation yields empirical gains: on CIFAR-10, GDD improves FID from 3.06 (Beta Diffusion) to 2.76 and reduces KID from 1.71×10⁻³ to 1.22×10⁻³, with higher sampling throughput at the same NFE. These improvements stem from modeling intra-group dependencies and the grouped KLUB objective, not just reparameterization.
> >
> > We agree that the Gamma–Dirichlet view makes the Beta-to-Dirichlet extension conceptually straightforward at the distributional level. Building on this foundation, our work contributes (i) a hierarchical, group-structured diffusion framework on high-dimensional simplices, (ii) an SDE and KLUB theory tailored to grouped Dirichlet noise, and (iii) an architecture that leverages these properties for image generation. We will revise the paper to present GDD more clearly as a principled generalization of Beta Diffusion (recovering it when $G=1$, $K=2$) while highlighting the elements that go beyond a mechanical extension.

---

### Author Response · Authors · 2025-12-01
**Summary for Area Chair**

Dear Area Chair,

Thank you and all the reviewers for your time and the valuable feedback on our submission. We have carefully considered all comments and have prepared a detailed rebuttal.

In light of the updated ICLR policy that prevents reviewers from further interacting during the rebuttal phase, I would like to provide this final summary to clarify our contributions, highlight the concrete improvements made during the response period, and ensure that our work is assessed with the most complete and accurate information.

Across the four reviews, there is clear consensus that the paper introduces a principled and well-motivated generative modeling framework based on grouped Dirichlet distributions, providing a meaningful structural prior for compositional data and effectively capturing intra-group dependencies that Gaussian or independent-noise models cannot. Reviewers consistently recognize the theoretical soundness of the approach—particularly the closed-form Dirichlet transitions and the stable KLUB objective—as well as the strong empirical results on CIFAR-10, where GDD outperforms Beta Diffusion in both FID/KID and efficiency. Multiple reviewers also highlight that GDD achieves faster training and sampling than traditional DDPM variants, indicating practical computational advantages in addition to improved sample quality.

We have addressed the reviewers’ key concerns as follows. For Reviewer 5TU9’s baseline-comparison critique, we have clarified and revised the relevant paper sections and provided more detailed explanations for the mathematical interpretation points they raised. In response to Reviewer fN7s’s request for additional experiments, we added an ablation study on the weight parameter $\omega$ and expanded the discussion of the ablations reported in Table 2. We also corrected the formatting and presentation issues noted by Reviewer 2RpY; Reviewer 2RpY has confirmed the revisions and indicated they will consider a higher score. We respectfully ask the Area Chair to take these substantive clarifications and updates into account when evaluating the submission.

In addition, Reviewer fN7s, Reviewer JJfZ, and Reviewer 2RpY all mentioned additional experiments related to grouping methods. In order to further demonstrate the effectiveness of the grouping method, we have added experiments.

In addition to grouping channels into semantically meaningful RGB feature sets, we further evaluated the generality of our grouped Dirichlet diffusion framework using two alternative grouping strategies: spatial pixel patching and random feature partitioning.
Experiments on CIFAR-10 (both qualitative samples and quantitative FID scores) show that spatial pixel patching yields reasonable performance when the image is divided into only a few coarse blocks by rules. However, its results remain slightly inferior to RGB grouping, as spatial partitioning disrupts the natural coherence of color distributions across the image. When the number of spatial patches increases, training and sampling become substantially slower while FID exhibits no meaningful improvement. When the division method becomes irregular, quantitative indicators will become worse.
In contrast, random feature partitioning severely breaks semantic structure: generated samples become visually incoherent, and quantitative metrics deteriorate sharply. This confirms that the model struggles to learn stable dependency patterns under arbitrary, non-semantic groupings. In summary, we have chosen to divide the color channels as the main explanatory content of the paper.

The additional experiments and corresponding explanations have been incorporated into the revised appendix. Figure 4 illustrates the two grouping strategies—RGB grouping and a four-patch spatial partition—while Table 6 reports the corresponding FID scores under varying NFEs. Figure 6 shows the generation effect of different grouping strategies on the CIFAR10.

We have updated the revised manuscript and added code supplementary materials for grouped experiments.

Thank you for your time and consideration.

Best regards,

The Authors of Submission[10689]

| Grouping Strategies | NFE=100 | NFE=200 | NFE=500 | NFE=1000 |
|:----:|:----:|:----:|:----:|:----:|
|Color channels partitioning |3.92|3.24|2.91|2.76|
|Spatial pixel patching(Regular Division)|4.18|3.53|3.01|2.94|
|Spatial pixel patching(Irregular Division)|328|298|288|273|
|Random feature partitioning|522|410|403|401|

---

### Meta-Review · Area_Chair_5Xxn · 2025-12-18

**Summary:**

Grouped Dirichlet Diffusion (GDD) extends Beta Diffusion to simplex-valued variables by introducing a user-specified group factorization, with closed-form forward transitions and a KL upper bound (KLUB) training objective. From the AC perspective, modeling the joint distribution as a product of group-specific distributions effectively treats groups as conditionally independent given the parameters, which is a fairly standard modeling choice and may not, by itself, justify a new named framework. In addition, parts of the notation are confusing. For example, Eq. 7 defines a Dirichlet distribution using a single vector of parameters as input, whereas Eq. 11 appears to define a Dirichlet distribution with two vector-valued inputs. It is unclear what this is meant to represent and, as written, the notation is confusing and non-standard.

**Reviewer Concerns:**

Reviews are mixed: two reviewers are marginally positive (scores 6), highlighting a clean core idea and competitive CIFAR-10 results, while two reviewers recommend rejection (scores 2), citing missing, weak, or unfair baselines and overstated empirical claims, as well as unclear presentation/notation and an under-validated motivation for “meaningful grouping.”

**Reviewer Scores:**

These negative ratings are unlikely to change, as their main concerns largely remain. These concerns include baseline fairness and positioning, limited novelty beyond a standard product-of-groups assumption, and insufficient clarity around compositional modeling.

---

### Decision · Program_Chairs · 2026-01-26

Reject